# UNDERSTANDING MULTIMODAL INSTRUCTION FORMAT FOR IN-CONTEXT LEARNING

## ABSTRACT

The field of vision and language machine learning has witnessed a surge in interest regarding in-context learning—a technique that enables rapid adaptation to new tasks with just a handful of annotated examples. To bolster the in-context learning capabilities of multimodal vision and language models, researchers have explored various instruction tuning formats. In this paper, we aim to study what should be the effective format for enhancing the in-context learning ability for vision and language models. We propose Unified Multimodal Instruction Tuning (UMIT), a framework to suggest how to construct a text-image interleaved instruction dataset by merging diverse visual instruction datasets in a unified multimodal instruction format. To examine the effectiveness of UMIT, we train several models based on OpenFlamingo in different multimodal instruction formats used by existing MLLMs. Extensive experiments confirm that UMIT can significantly improve the in-context learning ability on a wide range of vision-language tasks, compared with prior formats, including MME Benchmark and SEED-Bench. Furthermore, we conduct a comprehensive study on the impact of different components in multimodal instruction formats on the in-context learning ability of MLLMs in 3 traditional vision-language tasks. The results indicate that UMIT successfully constrains the model to focus on task-specific information within in-context exemplars by incorporating a task definition component, thus giving it remarkable advantages over prior formats on zero- and few-shot generalization during both the training and testing stages.

## 1 INTRODUCTION

GPT-4 (OpenAI, 2023) exhibits astounding performance in vision-language tasks. One charming property of GPT-4 is its' in-context learning ability, which enables rapid adaption to new tasks with just a handful of annotated examples. Although there are many Multimodal Large Language Models (MLLMs) proposed (Liu et al., 2023b; Zhu et al., 2023; Huang et al., 2023; Dai et al., 2023), they primarily focus on improving the zero-shot performance, leaving the way to improve in-context learning less explored.

In the research line of Multimodal in-context learning, Flamingo (Alayrac et al., 2022) and Open-Flamingo (Awadalla et al., 2023) are pioneering to achieve the goal by constructing large-scale multi-modal datasets and using the data for upstream pretraining. To further adapt to downstream tasks effectively with better in-context learning ability, *Otter* (Li et al., 2023b), an instruction tuning-based method, has been proposed based on OpenFlamingo. These methods involve creating diverse downstream text-image interleaved datasets and integrating them with the multimodal instruction format. To achieve better performance, researchers have been mainly dedicated to dataset construction for instruction tuning, where they show that increasing the dataset size, complexity, and diversity can consistently improve the model performance (Li et al., 2023a), leaving the importance of the multimodal instruction format less explored.

Due to the selection of different tasks for constructing visual instruction datasets, existing methods employ various multimodal instruction formats mainly composed of four components, including in-context exemplars, instruction, question and answer (shown in Table 1). For example, *Qwen-VL* (Bai et al., 2023) collects traditional vision-language tasks by simply concatenating samples from the same

task without any instructions to form the instruction format. Thus, training in this format is essentially performing multi-task learning while instruction tuning, limiting their in-context learning ability for fast adaptation. *Otter* (Li et al., 2023b) and *MMICL* (Zhao et al., 2023b) improved the process by applying an instance-level instruction format to merge the interleaved datasets for instruction tuning. Despite their success, we argue that the existing instruction-level instruction format has its limitations. One limitation is the inherent lack of task-level contextual understanding. This deficiency poses challenges for the model in comprehending the broader context of the questions it encounters, hindering its ability to swiftly adapt to diverse task domains and provide accurate responses. For instance, *LLaVA* (Liu et al., 2023b) introduces task-level system messages to prompt GPT-4 to accurately generate corresponding visual instruction datasets. Furthermore, it's worth noting that these models are fine-tuned based on foundation models (e.g., OpenFlamingo), which lack instance-level instructions during their pretraining phase. This absence of instance-level instruction introduces a discrepancy between the fine-tuning and pretraining stages, thereby potentially diminishing the efficiency of instruction tuning. Thus, a natural question emerges: "*What should be the multi-modal instruction format for instruction tuning?*" Furthermore, the multi-modal instruction format is the way to merge the collected image-text interleaved data for fine-tuning. In contrast to increasing the data diversity for enhancing the in-context learning ability, is it possible to use less data but with a more effective multi-modal instruction format to improve the model's in-context learning ability?

In this work, we present **U**nified **M**ultimodal **I**nstruction **T**uning (UMIT), a general framework to construct a text-image interleaved instruction dataset by merging diverse vision-language tasks in a unified multimodal instruction format. UMIT introduces an essential component to construct the multimodal instruction format, namely "task definition" in addition to in-context exemplars, instance-level instruction, question, and answer and is placed at the forefront. The goal of the task definition is to inform the model about the contextual task-level information and reduce the gap between the format inconsistency between the pretraining and fine-tuning stages. In addition, the inclusion of a task definition can lead to synergistic benefits by leveraging the efforts invested in data construction. For data construction, the main efforts in the literature aim to construct more diverse, complicated, and large datasets. Implicitly, this construction contributes to enhancing task diversity and naturally provides a task definition. As a result, our instruction format effectively harnesses this data, offering the potential to use less data for better performance.

UMIT consists of two key steps. It first collects a set of text-image interleaved datasets and then transfers them into the unified multimodal instruction formation for finetuning. We collect data from existing instruction datasets (Xu et al., 2022; Liu et al., 2023b; Gong et al., 2023; Li et al., 2023a; Zhao et al., 2023a), which naturally provide the task descriptions. To evaluate the potential of our method, we only randomly select a few examples for each task. Given the data, we then can convert it into our instruction format. To construct the task definition, since annotations are from different annotators, there exist significant differences in the instructions for different tasks. To reduce the impact of the format inconsistency on combining diverse tasks, we leverage an Oracle model (i.e., OpenAI GPT-3.5[1]) to transfer the multimodal instructions of diverse tasks in a unified style. To construct the in-context exemplars, we employ a retrieval-based approach. The rest of the components all match the corresponding fields in our unified format, so we directly follow their original styles and fill them into our format.

We conduct comprehensive experiments to evaluate the impact of different multimodal instruction formats on the in-context learning ability of MLLMs (§3.2). Specifically, although we use less data (150k, almost 20 times less than one baseline (*otter*)), our unified format demonstrates average improvements of 4.7 and 9.4 on the MME Benchmark (Fu et al., 2023) and SEED-Bench (Li et al., 2023c), respectively, compared to baselines in the in-context learning setting. Furthermore, We also perform a comprehensive study on three traditional vision-language tasks, analyzing the influence of the components in different formats on in-context learning ability (§3.3). Extensive experimental results show that UMIT significantly mitigates the loss of in-context learning capability caused by instruction tuning compared with prior formats, which means it can successfully combine various components into a more effective unified format. For further analysis, we also explore the effects of our proposed unified format on zero-shot performance (§3.4). The results show that our method effectively merges instruction datasets to enhance task diversity without degrading the zero-shot performance.

---

[1]https://platform.openai.com/docs/models/gpt-3-5

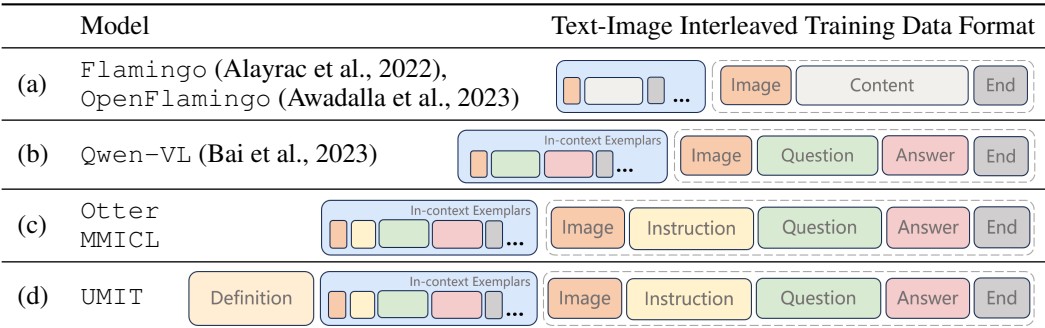

Table 1: The existing text-image interleaved instruction formats exhibit variations across different datasets for preserving in-context learning ability. For example, Qwen-VL and Otter (Li et al., 2023b) focus on preserving in-context learning ability without explicit instructions. Moreover, the majority of Otter's datasets are derived from real-world tasks, which means that its questions can be considered a form of instance-level instruction (light yellow). MMICL (Zhao et al., 2023b) incorporates diverse VL tasks and creates instance-level instruction templates for their instances. Distinct from these formats, we propose Unified Multimodal Instruction Tuning (UMIT) to unify different formats in various datasets while effectively expanding task diversity and preserving in-context learning ability.

## 2 UNIFIED MULTIMODAL INSTRUCTION TUNING

In this section, we introduce the principal components constituting the existing multimodal instruction formats, along with the recent research advancements as depicted in Table 1. We then introduce the pipeline we used to collect more diverse tasks from different datasets. Then we present Unified Multimodal Instruction Tuning (UMIT), an attempt to unify multimodal instruction format for preserving in-context learning capabilities while also assisting us in merging various visual instruction datasets to enhance task diversity.

### 2.1 PRELIMINARIES ON MULTIMODAL INSTRUCTION FORMAT

Flamingo (Alayrac et al., 2022) and OpenFlamingo (Awadalla et al., 2023) emerge as pioneering frameworks to address a diverse spectrum of vision-language tasks, encompassing visual question-answering, captioning, and image classification. Initially, the design exclusively incorporated the **exemplars** to bolster in-context learning. Formally, given a new instance $i$, the format $\mathcal{F}^i$ employed by Flamingo and OpenFlamingo is expressed as:

$$\mathcal{F}_1^i = ( \ [\mathbf{X}_e^1, \cdots, \mathbf{X}_e^N], \ \mathbf{X}_v^i, \ \mathbf{X}_l^i), \tag{1}$$

wherein, for instance $i$, in-context exemplars $\mathbf{X}_e^j$ are randomly selected to construct few-shot templates, while $\mathbf{X}_v^i$ and $\mathbf{X}_l^i$ denote the image and language content of instance $i$, respectively.

As delineated in Table 1, Qwen-VL (Bai et al., 2023) took the initiative to extend the framework to encompass three components. It notably segregates the language input into **question** and **answer** for the current instance, denoted as $\mathbf{X}_q^i$ and $\mathbf{X}_a^i$, respectively. The overall format thus evolves to:

$$\mathcal{F}_1^i = ( \ [\mathbf{X}_e^1, \cdots, \mathbf{X}_e^N], \ \mathbf{X}_v^i, \ \mathbf{X}_q^i, \ \mathbf{X}_a^i \ ). \tag{2}$$

Moreover, unlike the format utilized by Qwen-VL, Otter (Li et al., 2023b) and MMICL (Zhao et al., 2023b) incorporated the instance-level instruction component into the prompt format design for in-context instruction tuning, thereby enriching the overall format to become:

$$\mathcal{F}_2^i = ( \ [\mathbf{X}_e^1, \cdots, \mathbf{X}_e^N], \ \mathbf{X}_v^i, \ \mathbf{X}_{\texttt{instruct}}^i, \ \mathbf{X}_q^i, \ \mathbf{X}_a^i \ ). \tag{3}$$

Therefore, the current state-of-the-art multimodal instruction formats encapsulate four components: 1) **examplars**, 2) **instruction**, 3) **question**, and 4) **answer** corresponding to the current instance or context. As elucidated in Section 1, prevailing literature predominantly focuses on instance-level instruction tuning, which is highly specific to each data sample. Nonetheless, a high-level task-level orientation has been conspicuously overlooked. As shown in Table 1 (d), we devise a new component, **task definition**, to bridge this gap in our UMIT, which will be elaborated in Section 2.3. Additionally,

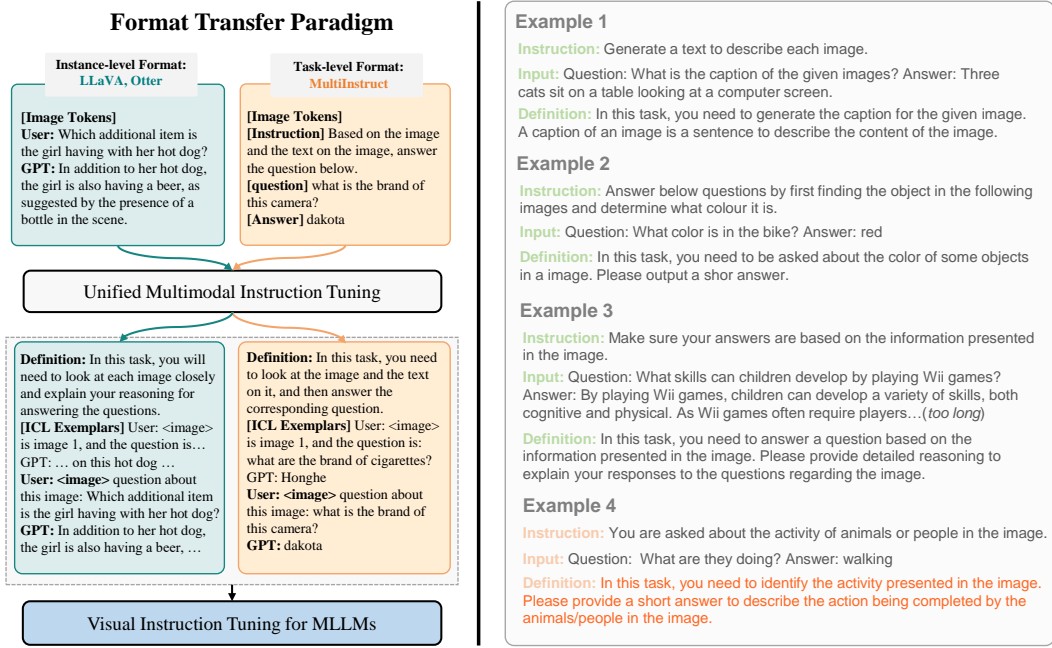

Figure 1: Our multimodal instruction format transfer paradigm and an example of generating a task definition. To obtain more accurate definitions, we incorporate task examples in our prompts.

we discern that the current exemplars are selected through various methods. We then further explore the influence of different exemplar retrieval methods on in-context learning, thus selecting the most effective exemplars to augment the performance further.

## 2.2 DATA COLLECTION PIPLINE FOR UMIT

As our task definition prompt and enhanced exemplar generation requires a prebuilt database, we first introduce the data collection pipeline. We conduct data collection and construction through the following three steps.

**Step i  Data Collection**   First, we gathered various datasets containing different tasks (55 in total), categorizing them into five main types, including image caption (IC), visual question answering (VQA), visual reasoning (VR), referential dialogue (RD), and language-only tasks. Detailed information about data resources can be found in Appendix A.

**Step ii  Format Transfer**   For the tasks within these different data resources, we organize their task-specific instructions and notice significant format discrepancies among them. Directly merging them would diminish the in-context learning ability on unseen tasks due to format inconsistency. Therefore, we transfer these instructions into a unified style of task definition.

**Step iii  Data Filiter**   During the training process, we employ filtering strategies described in Appendix C to further reduce the effects of increasing dataset scale while preserving the benefits of increasing task diversity.

## 2.3 TASK DEFINITION AS A NOVEL COMPONENT IN INSTRUCTION TUNING

Foundation models, pre-trained on massive data harvested from a multitude of sources, have exhibited formidable capabilities in generating insightful responses. Yet, there remains a margin for enhancement in terms of accuracy and professionalism. Various methods of instruction tuning have been proffered as a means to steer these models towards more precise outcomes. However, a notable limitation lies in the fact that such instructions tend to be instance-specific, as opposed to being driven by the overarching task at hand. This narrow focus could potentially hamstring the model's ability to generalize across a broader spectrum of queries, thereby underscoring the necessity for more holistic, task-oriented tuning methodologies. Adopting a more task-centric approach may pave the way for more adept handling of queries, propelling the model towards generating answers that are not only accurate but are imbued with a higher degree of professionalism.

Inspired by the above observations, we propose to improve the instruction tuning by introducing our **task definition** in UMIT. Denote the original task-level instruction as $\mathcal{I}$. We aim to transfer $\mathcal{I}$ of any style to a more unified task definition $\mathcal{D}$, to constrain learning task-specific information contained in the in-context exemplars. Inspired by Liang et al. (2023), we employ GPT-3.5 (text-davinci-003) as the Oracle model $f_{oracle}$ to transfer different styles of task-level instructions into task definitions with a unified style for each task via in-context learning. Specifically, $k$ seed instruction-examplar pairs $\{(\mathbf{X}_{\text{instruct}}^1, \mathbf{X}_{\text{e}}^1), \ldots, (\mathbf{X}_{\text{instruct}}^k, \mathbf{X}_{\text{e}}^k)\}$ are manually selected from several tasks. Then we craft a task definition $\mathcal{D}_j$ for each instruction-exemplar pair $(\mathbf{X}_{\text{instruct}}^j, \mathbf{X}_{\text{e}}^j)$, and consequently get a set of instruction-exemplar-definition pairs $\{(\mathbf{X}_{\text{instruct}}^1, \mathbf{X}_{\text{e}}^1, \mathcal{D}_1), \ldots, (\mathbf{X}_{\text{instruct}}^k, \mathbf{X}_{\text{e}}^k, \mathcal{D}_k)\}$, which is used as the prompt for the Oracle model. Then, given a new instruction-exemplar pair $(\mathbf{X}_{\text{instruct}}^{\text{new}}, \mathbf{X}_{\text{e}}^{\text{new}})$, the Oracle model will generate the corresponding unified-style task definition $\mathcal{D}_{\text{new}}$ as follows:

$$\mathcal{D}_{\text{new}} = f_{oracle}\left((\mathbf{X}_{\text{instruct}}^1, \mathbf{X}_{\text{e}}^1, \mathcal{D}_1), \ldots, (\mathbf{X}_{\text{instruct}}^k, \mathbf{X}_{\text{e}}^k, \mathcal{D}_k), (\mathbf{X}_{\text{instruct}}^{\text{new}}, \mathbf{X}_{\text{e}}^{\text{new}})\right). \quad (4)$$

As shown in Tabel 1 (right), we assist GPT-3.5 in better understanding task types by incorporating task instances, such as distinguishing between tasks that require long answers (e.g., LLaVA-instruction-150k (Liu et al., 2023b)) and those that require short answers (e.g., GQA (Hudson & Manning, 2019)). However, we contend that creating a robust and meaningful definition for each instance from real-world scenarios is an ongoing challenge that requires further exploration in the future.

## 2.4 ENHANCED EXEMPLARS SAMPLING TO BOOST IN-CONTEXT LEARNING

Prior works in multimodal domains (Yang et al., 2022; Li et al., 2023b; Bai et al., 2023; Zhao et al., 2023b) have shown that adding in-context exemplars can preserve the model's in-context learning capabilities. However, the impact of different in-context exemplar selection approaches on in-context learning ability remains an under-explored facet. Therefore, in this section, we comprehensively investigate the selection of in-context exemplars using both image and text features. Specifically, We define $C = \{T_1, T_2, \ldots, T_N\}$ as the retrieval database, which is a collection of $N$ vision-language tasks, where each task $T = \{(\mathbf{X}_{\text{v}}^1, \mathbf{X}_{\text{q}}^1, \mathbf{X}_{\text{a}}^1), \ldots, (\mathbf{X}_{\text{v}}^n, \mathbf{X}_{\text{q}}^n, \mathbf{X}_{\text{a}}^n)\}$ is a dataset consisting of $n$ image-question-answer pairs collected in Section 2.2.

For data retrieval, we denote $\mathbf{Z}_{\text{e}} = (\mathbf{X}_{\text{v}}, \mathbf{X}_{\text{q}}, \mathbf{X}_{\text{a}})$ as a retrieval exemplar, and use an image encoder $\mathbf{E}_{\text{Image}}$ and a text encoder $\mathbf{E}_{\text{Text}}$, mapping a text or image instance to a $d$-dimensional latent space. Then, we select in-context exemplars based on the cosine similarity of their representations. Specifically, we initialize the image encoder with CLIP-ViT (Radford et al., 2021), a pretrained vision transformer, and we employ text-embedding-ada-002 [2] as the text encoder. Therefore, for a new exemplar $\mathbf{Z}_{\text{e}}^{\text{new}} = (\mathbf{X}_{\text{v}}^{\text{new}}, \mathbf{X}_{\text{q}}^{\text{new}}, \mathbf{X}_{\text{a}}^{\text{new}})$ belonging to task $T$, we respectively searches $k$ nearest samples in both text-encoded and image-encoded latent space, according to the cosine similarity. We denote this retrieval pipeline as $\mathcal{R}$:

$$\mathcal{R}(\mathbf{Z}_{\text{e}}^{\text{new}}) = \{\underset{\mathbf{Z}_{\text{e}}}{\text{Top}}\, k\left(\cos(\mathbf{E}_{\text{Image}}(\mathbf{Z}_{\text{e}}^{\text{new}}), \mathbf{E}_{\text{Image}}(\mathbf{Z}_{\text{e}}))\right), \quad (5)$$

$$\underset{\mathbf{Z}_{\text{e}}}{\text{Top}}\, k\left(\cos(\mathbf{E}_{\text{Text}}(\mathbf{Z}_{\text{e}}^{\text{new}}), \mathbf{E}_{\text{Text}}(\mathbf{Z}_{\text{e}}))\right)\} \quad (6)$$

$$= \{\hat{\mathbf{Z}}_{\text{e}}^1, \ldots, \hat{\mathbf{Z}}_{\text{e}}^{2k}\}. \quad (7)$$

Where $\mathbf{E}_{\text{Text}}$ use the $\mathbf{X}_{\text{q}}$ and $\mathbf{X}_{\text{a}}$ to compute the text embedding during the training stage.

## 2.5 UNIFIED MULTIMODAL INSTRUCTION FORMAT

Based on the diverse tasks collected as mentioned above, along with specific task definitions and in-context exemplars, we propose a unified multimodal instruction format $\mathcal{F}_3$:

$$\mathcal{F}_3 = (\mathcal{D}_{\text{new}}, [\mathbf{X}_{\text{e}}^1, \ldots, \mathbf{X}_{\text{e}}^N], \mathbf{X}_{\text{v}}^{\text{new}}, \mathbf{X}_{\text{instruct}}^{\text{new}}, \mathbf{X}_{\text{q}}^{\text{new}}, \mathbf{X}_{\text{a}}^{\text{new}}), \quad (8)$$

Where $\mathcal{D}_{\text{new}}$ denotes the unified task definition generated by prompting the Oracle model as Eq. 4. Note that for each instance $j$, the in-context exemplar consists of $(\hat{\mathbf{X}}_{\text{v}}^j, \mathbf{X}_{\text{instruct}}^i, \hat{\mathbf{X}}_{\text{q}}^j, \hat{\mathbf{X}}_{\text{a}}^j)$, where $(\hat{\mathbf{X}}_{\text{v}}^j, \hat{\mathbf{X}}_{\text{q}}^j, \hat{\mathbf{X}}_{\text{a}}^j) = \mathcal{R}((\mathbf{X}_{\text{v}}^{\text{new}}, \mathbf{X}_{\text{q}}^{\text{new}}, \mathbf{X}_{\text{a}}^{\text{new}}))_j$ is the $j$-th retrieved exemplar.

---

[2]https://platform.openai.com/docs/guides/embeddings/what-are-embeddings

| Models | #Exp | Shots | Existen. | Count | Pos. | Color | Poster | Cele. | Scene | Land. | Art. | OCR | Perception |
|---|---|---|---|---|---|---|---|---|---|---|---|---|---|
| OpenFlamingo[†] | - | 0 | 103.33 | 43.33 | 60.00 | 55.00 | 111.56 | 91.76 | 95.25 | 54.75 | 78.50 | 55.00 | 748.50 |
| OpenFlamingo[†] | - | 2 | 150.00 | 58.33 | 63.33 | 75.00 | 89.46 | 56.18 | 108.50 | 50.00 | 79.25 | 70.00 | 800.05 |
| OpenFlamingo[†] | - | 4 | 145.00 | 55.00 | 63.33 | 85.00 | 84.35 | 60.59 | 125.50 | 50.00 | 94.50 | 95.00 | 858.28 |
| Otter[†] | 2.8M | 0 | 185.00 | 88.33 | 65.00 | 113.33 | 117.69 | 110.29 | 162.25 | 99.50 | 105.25 | 50.00 | 1096.65 |
| Otter[†] | 2.8M | 2 | **190.00** | **125.00** | 55.00 | 75.00 | 119.39 | 121.47 | 159.25 | 105.25 | 109.00 | 57.50 | 1116.86 |
| Otter[†] | 2.8M | 4 | **190.00** | 100.00 | 60.00 | 85.00 | 113.27 | 115.88 | 149.75 | 105.25 | 96.50 | 72.50 | 1088.15 |
| OpenFlamingo ($\mathcal{F}_1$) | 148k | 0 | 165.00 | 78.33 | 56.67 | 125.00 | 120.07 | 105.29 | 140.75 | 94.50 | 72.25 | 72.50 | 1030.36 |
| OpenFlamingo ($\mathcal{F}_1$) | 148k | 2 | 175.00 | 91.67 | 63.33 | 125.00 | 150.00 | 125.88 | 158.75 | 123.00 | 113.25 | 80.00 | 1205.88 |
| OpenFlamingo ($\mathcal{F}_1$) | 148k | 4 | 175.00 | 83.33 | 60.00 | 125.00 | **158.16** | 135.88 | 157.75 | 122.25 | 109.75 | 80.00 | 1207.13 |
| Otter ($\mathcal{F}_2$) | 148k | 0 | 180.00 | 100.00 | 75.00 | 120.00 | 81.63 | 52.65 | 131.75 | 59.00 | 72.25 | 50.00 | 922.28 |
| Otter ($\mathcal{F}_2$) | 148k | 2 | 180.00 | 73.33 | 75.00 | 108.33 | 147.28 | 162.94 | **168.25** | **147.75** | 99.50 | 80.00 | 1242.39 |
| Otter ($\mathcal{F}_2$) | 148k | 4 | 180.00 | 48.33 | 56.67 | 100.00 | 150.00 | 127.65 | 166.75 | 144.00 | 84.25 | 80.00 | 1137.65 |
| UMIT ($\mathcal{F}_3$) | 148k | 0 | 180.00 | 53.33 | 48.33 | 103.33 | 138.10 | 129.41 | 157.25 | 126.00 | 95.00 | 65.00 | 1095.76 |
| UMIT ($\mathcal{F}_3$) | 148k | 2 | 180.00 | 108.33 | **90.00** | 118.33 | 148.30 | 161.18 | 160.75 | 145.50 | **122.75** | 102.50 | **1337.64** |
| UMIT ($\mathcal{F}_3$) | 148k | 4 | 185.00 | 98.33 | 70.00 | **130.00** | 154.42 | **163.82** | 161.50 | 146.50 | 116.00 | **102.50** | 1328.08 |

Table 2: Zero- and Few-shot evaluation of coarse-grained and fine-grained recognition and OCR on MME Benchmark (Fu et al., 2023). Models with † refer to our measure while indicating the results reported from their paper. Otter ($\mathcal{F}_2$) refers that we instruction fine-tune OpenFlamingo by employing the format used by Otter sytle ($\mathcal{F}_2$) mentioned in Equation 3.

## 3 EXPERIMENTS

### 3.1 GENERAL SETUP

**Format and Baselines.** These instruction formats primarily consist of five components, as referenced in Section 2.1: (1) task definition ($\mathbf{D}$), (2) in-context exemplars ($\mathbf{E}$), (3) instruction ($\mathbf{I}$) that is customized for each individual example, (4) Question of the current instance ($\mathbf{Q}$), (5) Answer of the current instance ($\mathbf{A}$). We can map these components into the existing baselines.

Since we mainly focus on the in-context learning ability, we mainly consider the methods following the directions of Flamingo as baselines, which include: **(1) OpenFlamingo** (Awadalla et al., 2023), which is an open-sourced Flamingo (Alayrac et al., 2022) with an internal LLM of MPT-7b. The instruction format is composed of $EQA$. **(2) Otter** (Li et al., 2023b) is trained on OpenFlamingo with MIMIC-IT datasets (including 2.8M samples) (Li et al., 2023a) and first introduces in-context learning into visual instruction tuning. The instruction format is composed of $EIQA$

**Hyper-parameter and Training Details.** We train UMIT on our proposed dataset in DEIQA format ($\mathcal{F}_3$). We utilize DeepSpeed (Rasley et al., 2020) for optimization during the training process. The AdamW (Loshchilov & Hutter, 2018) optimizer is used, with $\beta_1 = 0.9, \beta_2 = 0.999$, and a weight decay of 0.01. All training runs on 4 NVIDIA A100 GPUs, with a total batch size of 128, a learning rate of $2 \times 10^{-5}$ for the second stage. The maximum sequence length is fixed at 2000 and BF16 precision is used for both training and inference. Detailed information can be found in Appendix C.

### 3.2 EVALUATION ON MULTI-MODAL BENCHMARKS

To evaluate the effectiveness of our method, here, we select two commonly employed benchmarks: MME(Fu et al., 2023) and SEED (Li et al., 2023c). As we aim to assess the in-context learning ability, we evaluate the performance by using $k$ example as an in-context example, where $k = \{0,2,4\}$. We compare our method with the officially released checkpoint. We denote the baseline as OpenFlamingo† and Otter†. Since the baseline methods are trained on different sizes of instruction data compared to ours, for a fair comparison, we also apply the baselines on our training corpus with their instruction tuning format. We denote them as OpenFlamingo($\mathcal{F}_1$) and Otter($\mathcal{F}_2$).

### 3.2.1 EXPERIMENT RESULTS ON MME BENCHMARK

We conducted comprehensive testing on the MME Benchmark (Fu et al., 2023) to assess the impact of different formats and task diversity on the zero- and few-shot capabilities of our models. The MME

| Models | #Exp | Shots | Scene | Identity | Attr. | Loc. | Count. | Spatial | Interac. | Reason. | Text Rec. | Avg. |
|---|---|---|---|---|---|---|---|---|---|---|---|---|
| OpenFlamingo[†] | - | 0 | 55.1 | 49.5 | 47.9 | 37.4 | 37.8 | 33.8 | 40.2 | 43.2 | 30.6 | 46.3 |
| OpenFlamingo[†] | - | 2 | 55.6 | 51.3 | 49.6 | 39.3 | 42.1 | 32.7 | 43.3 | 45.3 | 44.7 | 48.2 |
| OpenFlamingo[†] | - | 4 | 56.9 | 51.8 | 50.1 | 38.6 | 43.7 | 32.7 | 44.3 | 43.8 | 45.9 | 48.9 |
| Otter[†] | 2.8M | 0 | 56.6 | 51.2 | 49.4 | 38.9 | 39.6 | 35.3 | 41.2 | 45.9 | 30.6 | 47.9 |
| Otter[†] | 2.8M | 2 | 55.3 | 51.2 | 49.8 | 39.3 | 42.7 | 32.9 | 45.4 | 46.5 | 44.7 | 48.3 |
| Otter[†] | 2.8M | 4 | 54.7 | 50.8 | 49.2 | 39.2 | 42.2 | 32.6 | 45.4 | 45.6 | 44.7 | 47.9 |
| OpenFlamingo ($\mathcal{F}_1$) | 148k | 0 | 55.9 | 54.2 | 57.8 | 42.9 | 47.6 | 39.0 | 48.5 | 45.3 | **48.2** | 52.9 |
| OpenFlamingo ($\mathcal{F}_1$) | 148k | 2 | 57.9 | 56.2 | 62.0 | 44.6 | 50.6 | 41.9 | 53.6 | 46.5 | 43.5 | 55.7 |
| OpenFlamingo ($\mathcal{F}_1$) | 148k | 4 | 58.0 | 55.5 | 61.4 | 43.3 | 51.5 | 40.2 | 51.6 | 47.1 | 38.8 | 55.4 |
| Otter ($\mathcal{F}_2$) | 148k | 0 | 61.2 | 56.6 | 62.0 | 45.3 | 50.0 | 42.5 | 52.6 | 49.2 | 28.2 | 56.5 |
| Otter ($\mathcal{F}_2$) | 148k | 2 | 59.3 | 55.6 | 62.0 | 45.7 | 53.3 | 42.3 | 54.6 | 46.8 | 28.2 | 56.4 |
| Otter ($\mathcal{F}_2$) | 148k | 4 | 60.0 | 56.2 | **62.4** | 45.9 | **53.8** | 42.8 | 54.6 | 47.7 | 28.2 | 57.0 |
| UMIT ($\mathcal{F}_3$) | 148k | 0 | **62.6** | 57.1 | 59.2 | **47.8** | 49.9 | 38.7 | **56.7** | 50.2 | 38.2 | 56.1 |
| UMIT ($\mathcal{F}_3$) | 148k | 2 | 61.5 | **58.0** | 61.9 | 47.0 | 52.5 | 42.8 | 53.6 | **52.6** | 30.6 | 57.3 |
| UMIT ($\mathcal{F}_3$) | 148k | 4 | 61.5 | 57.6 | 62.0 | 45.3 | 53.3 | **43.1** | 50.5 | 50.5 | 36.5 | **57.3** |

Table 3: Zero- and Few-shot evaluation on SEED-Bench (Li et al., 2023c) consists of 19K multiple-choice questions with accurate human annotations, covering 12 evaluation dimensions including both the spatial and temporal understanding.

Benchmark is a comprehensive multimodal benchmark used to evaluate the abilities of MLLMs across 14 tasks. It can be divided into perception and cognition benchmarks. Each task in the cognition benchmark contains only 20 examples, which can result in high variance in the evaluation of different checkpoints. Therefore, we only evaluate our models on the perception benchmark, which consists of 10 tasks: existence, count, position, color, posters, celebrities, scenes, landmarks, and artworks.

Specifically, as shown in Table 2, we can find that: (1) Compared with other MLLMs that also emphasize in-context learning, such as OpenFlamingo (Awadalla et al., 2023) and Otter (Li et al., 2023b), `UMIT`, trained on datasets merged through our `UMIT`, has demonstrated superior performance in terms of in-context learning capabilities. (2) Due to the training dataset used by Otter being predominantly composed of real-world VL tasks while lacking traditional VL tasks (Li et al., 2023a), `UMIT`, which is trained on a more diverse dataset, manages to attain a comparable performance while utilizing only 5% of its data volume. (3) For a fair comparison between different multimodal instruction formats, we also introduce OpenFlamingo ($\mathcal{F}_1$) and Otter ($\mathcal{F}_2$), which are trained on our dataset while preserving their own instruction formats. The results again demonstrate that `UMIT` can improve both zero- and few-shot task generalization performance by unifying the multimodal instruction format.

### 3.2.2 EXPERIMENT RESULTS ON SEED-BENCH

We further evaluate the impact of different instruction formats on SEED-Bench (Li et al., 2023c), a comprehensive benchmark consisting of 19K multiple choice questions with accurate human annotations including both spatial and temporal comprehension. We choose the spatial benchmark for our evaluation, which includes 9 tasks (number of included examples): Scene Understanding (3158), Instance Identity (1831), Instance Attribute (4649), Instance Location (978), Instance Counting (2447), Spatial Relation (657), Instance Interaction (97), Visual Reasoning (331), and Text Recognition (85).

Results are shown in Table 3, and we obtain conclusions similar to those on the MME benchmark: (1) in-context learning: `UMIT` outperforms all existing OpenFlamingo-based MLLMs in the 4-shot setting and also surpasses two other models trained on existing formats used to construct text-image interleaved datasets. (2) task diversity: Compared to OpenFlamingo and Otter, `UMIT` exhibits a significant advantage in both zero- and few-shot performance. This suggests that merging different datasets by unifying the instruction format does indeed effectively increase task diversity.

### 3.3 ABLATION STUDY

**Effects of multimodal instruction formats on in-context learning.** As shown in Table 4, we evaluate a total of three different multimodal instruction formats, including `EQA`, `EIQA`, `DEIQA`,

| Data Format | Retrieval Method | HatefulMemes (Kiela et al., 2020) | | | | VizWiz (Gurari et al., 2018) | | | | ISEKAI (Tai et al., 2023) | | | |
|---|---|---|---|---|---|---|---|---|---|---|---|---|---|
| | | $k=0$ | $k=2$ | $k=4$ | $k=8$ | $k=0$ | $k=2$ | $k=4$ | $k=8$ | $k=0$ | $k=4$ | $k=8$ | $k=16$ |
| EQA ($\mathcal{F}_1$) | mixed | 51.40 | 50.70 | 54.00 | 51.30 | 11.91 | 16.54 | 28.91 | **45.71** | 0.18 | 0.48 | 0.57 | 0.54 |
| EIQA ($\mathcal{F}_2$) | mixed | 54.44 | 54.57 | 53.89 | 50.56 | 24.74 | 28.93 | 29.38 | 30.90 | 0.00 | 0.38 | 0.46 | 0.58 |
| EIQA ($\mathcal{F}_2$) | image | 57.79 | 53.67 | 51.94 | 51.98 | 22.99 | 28.95 | 30.10 | 30.16 | 0.01 | 0.40 | 0.48 | 0.50 |
| EIQA ($\mathcal{F}_2$) | text | 56.76 | 55.10 | 56.96 | 55.52 | 17.13 | 27.58 | 28.27 | 30.01 | 0.00 | 0.37 | 0.51 | 0.53 |
| Testing-time Format Transfer | | | | | | | | | | | | | |
| DEIQA ($\mathcal{F}_3$) | image | 55.11 | 54.58 | 54.28 | 54.36 | 26.13 | 32.69 | 33.66 | 35.42 | 0.25 | 0.42 | 0.51 | 0.57 |
| DEIQA ($\mathcal{F}_3$) | text | 59.38 | **56.66** | 57.80 | 54.94 | 26.54 | 31.68 | 31.40 | 32.28 | 0.00 | 0.41 | 0.54 | 0.57 |
| Training time Format Transfer | | | | | | | | | | | | | |
| DEIQA ($\mathcal{F}_3$) | random | 61.37 | 54.15 | 52.49 | 50.61 | 27.15 | 29.53 | 30.66 | 31.21 | 0.18 | 0.47 | 0.56 | 0.64 |
| DEIQA ($\mathcal{F}_3$) | image | 59.09 | 55.57 | 55.94 | 56.88 | **27.45** | 32.69 | 33.75 | 34.95 | **0.32** | **0.51** | **0.62** | 0.67 |
| DEIQA ($\mathcal{F}_3$) | text | 57.96 | 55.69 | **60.19** | **59.18** | 26.79 | **33.55** | **36.06** | 37.60 | 0.22 | 0.47 | 0.57 | 0.65 |
| DEIQA ($\mathcal{F}_3$) | mixed | **61.86** | 54.65 | 55.78 | 51.77 | 27.03 | 29.66 | 31.02 | 31.87 | 0.20 | 0.41 | 0.61 | **0.84** |

Table 4: Comparison of different multimodal instruction formats on few-shot evaluation. $k$ is the number of in-context examples that are randomly selected in the inference stage. EQA, EIQA, and DEIQA respectively represent the ways of constructing text-image interleaved data based on OpenFlamingo ($\mathcal{F}_1$), Otter ($\mathcal{F}_2$), and Ocotpus ($\mathcal{F}_3$). "Testing-time Format Transfer" refers that we train the model in format ($\mathcal{F}_2$) while testing the model in format ($\mathcal{F}_3$). "Training time Format Transfer" means that we train and test the model both in format ($\mathcal{F}_3$).

on three vision-language tasks (HatefulMemes (Kiela et al., 2020), VizWiz (Gurari et al., 2018), ISEKAI (Tai et al., 2023)).

The results demonstrate that during both test and training times, DEIQA is the most effective format for improving in-context learning. Firstly, by transferring EQA format to EIQA format, we observe a decrease in in-context learning ability. In the highest shot setting for the two tasks, there is a decrease of 0.74 and 14.81, respectively. We contend that this is due to the incorporation of instructions, which widens the gap between visual instruction tuning and pertaining in OpenFlamingo. Secondly, we train on the EIQA format but transfer the test samples to the DEIQA format at test time. In the highest shot setting, there is an improvement of 2.38, 5.26, and 0.07 for each of the three tasks, respectively. Finally, when we employ the DEIQA format during both training and testing stages, there is a significant improvement compared to the EQA and EIQA baselines in all settings except for the 8-shot setting in VizWiz task. We argue that in-context learning aims to teach the model to respond to "unanswerable" for blurry pictures in the VizWiz task. However, current MLLMs tend to answer "yes" due to data biases, which leads to a decrease in performance on this task.

**Select in-context exemplars based on different features.** We also conduct a study about the retrieval method using different features (including image and text). random means we randomly select in-context exemplars to form the DEIQA format, while image and text represent the in-context exemplars selected by image and text features, respectively (Section 2.4). And mixed includes in-context exemplars in image and text. Based on the results in Table 4, our conclusion is: The unified format composed of in-context exemplars retrieved through different features has a different impact on in-context learning in various downstream tasks. For example, in the HatefulMemes task, the model needs to assess whether the text in the meme is hateful, and this text is provided in the instruction, which means instruction is the more important part of in-context. Therefore, we observe that a unified format composed of text in-context exemplars is more helpful for in-context learning ability on this task.

## 3.4 FURTHER ANALYSIS

### 3.4.1 ENHANCING TASK DIVERSITY WITH THE UNIFIED FORMAT.

To explore whether the UMIT can effectively merge different tasks by unifying the instruction format to enhance task diversity, we design a set of experiments. Specifically, we first randomly select 8 VQA tasks from our collection of 55 tasks (denoted as vqa). Then We choose another 8 different VQA tasks, denoted as same and 8 tasks distinct from VQA, denoted as diff. By freely combining

| Dataset | Shots | vqa+ same | vqa+ diff | all |
|---|---|---|---|---|
| Hateful Memes | 0 | 55.26 | 57.11 | **58.37** |
| | 4 | 54.16 | 55.44 | **55.49** |
| VizWiz (val) | 0 | 25.21 | 24.00 | **25.61** |
| | 4 | 34.35 | 33.76 | **34.58** |
| ISEKAI | 0 | 0.18 | 0.18 | **0.19** |
| | 4 | 0.39 | 0.42 | **0.47** |
| Flickr30K | 0 | 61.03 | 67.29 | **68.22** |
| | 4 | 67.61 | 70.33 | **71.96** |

Table 5: Experiments for enhancing task diversity. We set the dataset to `vqa+same`, `vqa+diff`, or `vqa+same+diff` in format ($\mathcal{F}_3$) and select the in-context exemplars through the `text` retrieval method.

| Models | Flickr 30K | SciQA image | OK-VQA | Text VQA |
|---|---|---|---|---|
| Flamingo-9B | 61.5 | - | **44.7** | 31.8 |
| OpenFlamingo-9B | 59.5 | - | 37.8 | 24.2 |
| Otter[†] | 63.6 | - | 42.3 | 27.2 |
| InstructBLIP-7B | **82.4** | 60.5 | - | - |
| **UMIT** | 70.2 | **60.9** | 43.3 | **33.0** |

Table 6: Zero-shot evaluation was conducted across multiple datasets, including Flickr30K, ScienceQA (Following Dai et al. (2023), we only evaluate on the set with image context), OK-VQA, and TextVQA. The results confirm that our UMIT does not have an impact on zero-shot performance.

these three groups of tasks using `UMIT`, we can obtain different training sets. The results are shown in Table 5, we can observe that `vqa+same+diff` achieve the best performance across different shot settings in all three tasks, suggesting that `UMIT` can indeed enhance zero- and few-shot performance by merging different tasks.

### 3.4.2 THE IMPACT OF DEIQA FORMAT ON THE ZERO-SHOT PERFORMANCE.

Table 6 shows the comparison between `UMIT` and some similar MLLMs on Flickr30K, ScienceQA, OK-VQA, and TextVQA tasks. The results suggest that `UMIT` is ahead of the MLLMs based on OpenFlamingo, so we believe that `UMIT` has minimal impact on zero-shot performance.

## 4 RELATED WORK

**Visual Instruction tuning.** Progress has been witnessed in transferring the powerful capabilities of instruction tuning to the multimodal domain. MultiInstruct (Xu et al., 2022) pioneered to collect a large number of vision-language (VL) tasks formatted in **task-level** instructions to improve zero-shot generalization of vision-language models on unseen tasks. With the emergence of multimodal large language models (MLLMs), recent works show interest in constructing high-quality visual instruction datasets with various instruction formats to unlock the potential of MLLMs on both zero- and few-shot generalization. Existing works (Liu et al., 2023b; Zhu et al., 2023; Dai et al., 2023; Chen et al., 2023b;a) prefers to adopt the instruction format that only contains a single image-text pair per sample (Text here generally includes instruction, question, answer), which leads to great zero-shot performance while being attributed to the lack of the in-context learning ability.
**Multimodal in-context learning.** Flamingo is the pioneering work to support in-context learning in the multi-modal domain by constructing MultiModal MassiveWeb(M2W) and employing the upstream training. Following this line of thought, the other works (Li et al., 2023b; Zhao et al., 2023b) focus on constructing text-image interleaved instruction datasets by adding related in-context exemplars, thus enhancing the instruction comprehension ability of MLLMs while preserving the in-context learning capacity.

## 5 CONCLUSION

In this paper, we propose the unified multimodal instruction tuning framework (`UMIT`) to merge diverse tasks with limited data by unifying instructions with different styles from various datasets, thus effectively enhancing in-context learning ability. The results show that our designed format has significant advantages over prior formats in in-context learning on various vision-language tasks and benchmarks. In general, we investigate a relatively unexplored facet, i.e., how to construct the text-image interleaved instruction datasets for preserving in-context learning ability, and we anticipate that our research could encourage further exploration in this area.

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

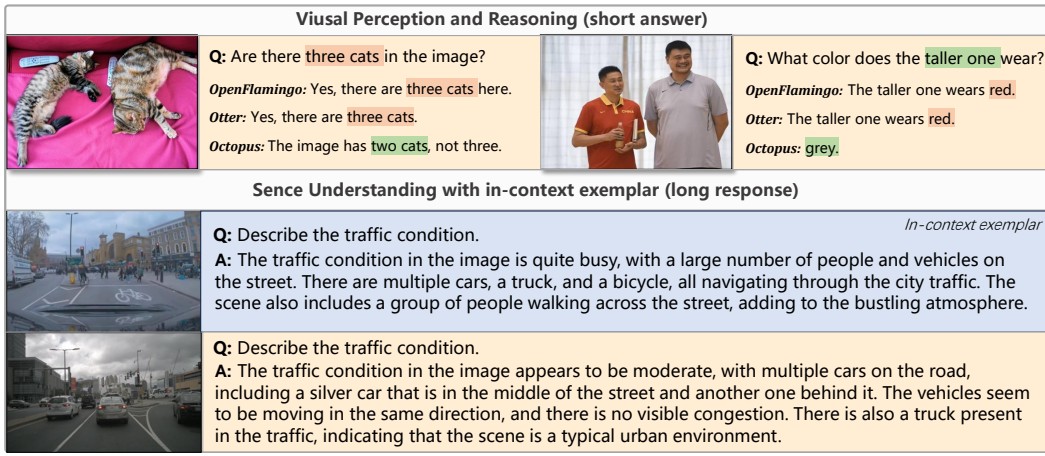

Figure 2: Examples illustrate that our model can avoid being misled in traditional vision-language tasks (short answers) while also demonstrating improved position awareness for visual reasoning. Additionally, our model can learn from exemplars to generate long responses (human preferences).

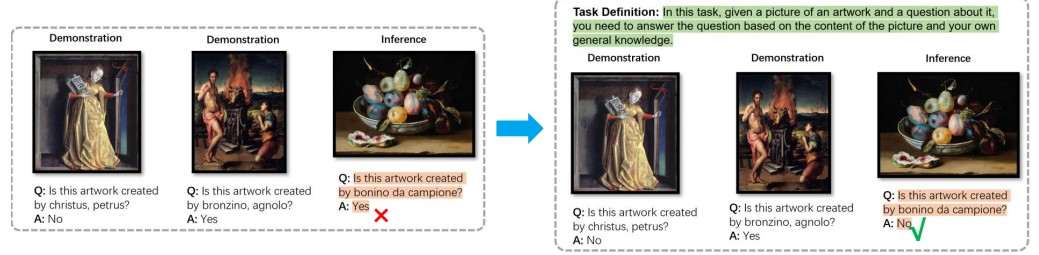

Figure 3: This example illustrates that by incorporating task definition, the model can learn to understand in-context exemplars, thereby enhancing in-context learning ability.

| Dataset | IC | VQA | VR | RD | Text | Few-Shot | # Tasks | # Exs |
|---------|----|----|----|----|------|----------|---------|-------|
| LLaVA[†] | ✔ | ✔ | ✔ | ✗ | ✗ | ✗ | 3 | 158k |
| MiniGPT-4[†] | ✔ | ✗ | ✗ | ✗ | ✗ | ✗ | N/A | 3.5k |
| Otter | ✔ | ✔ | ✔ | ✗ | ✗ | ✔ | 47 | ∼ 2.8M |
| MultiModal-GPT | ✔ | ✔ | ✔ | ✗ | ✔ | ✗ | 7 | ∼ 164k |
| InstructBLIP | ✔ | ✔ | ✔ | ✗ | ✗ | ✗ | 13 | ∼ 1.6M |
| M³IT | ✔ | ✔ | ✔ | ✔ | ✗ | ✗ | 40 | 2.4M |
| Clever Flamingo[†] | ✔ | ✔ | ✔ | ✔ | ✗ | ✗ | 37 | 975k |
| Shikra[†] | ✔ | ✔ | ✔ | ✔ | ✗ | ✗ | 4 | ∼ 90k |
| Emu[†] | ✔ | ✔ | ✔ | ✗ | ✔ | ✗ | 4 | ∼ 282k |
| UMIT[†] | ✔ | ✔ | ✔ | ✔ | ✔ | ✔ | 55 | 148k |

Table 7: Summary of visual instruction tuning datasets used by existing models. [†] indicates that the model is trained in two stages.

# A  VISION-LANGUAGE TASK COLLECTION

Previous works have demonstrated that extending visual instruction datasets can improve performance on unseen tasks with different multimodal instruction formats (Xu et al., 2022; Dai et al., 2023; Li et al., 2023b; Zhao et al., 2023b), as shown in Table 1. Therefore, we select 55 tasks from these datasets and merge them into a unified multimodal instruction format for training and evaluation.

## A.1  DATASET RESOURCE

**Image Caption (IC)**  The aim of this type of image caption task is to evaluate the model's overall perception of images, which is the foundation of all vision-language tasks. For example, we include MSCOCO (Lin et al., 2014) and Flickr30K (Young et al., 2014) datasets for short image descriptions, but LLaVA-Instruct-23k (Liu et al., 2023b) and SVIT (Zhao et al., 2023a) datasets for detailed image descriptions.

**Visual Question Answering (VQA)**  The majority of vision-language tasks can be generally viewed as Visual Question Answering (VQA) tasks, requiring the model to provide answers to queries related to the image. Therefore, we collect 17 VQA tasks from diverse datasets, such as VQAv2 (Goyal et al., 2017), OK-VQA (Marino et al., 2019), GQA (Hudson & Manning, 2019), OCR-VQA (Mishra et al., 2019), and TDIUC (Kafle & Kanan, 2017), among others.

**Visual Reasoning (VR)**  This category of task serves to assess the model's proficiency in visual reasoning. Specifically, we include some reasoning tasks: RD data generated by Chen et al. (2023b), Visual Reasoning in Natural Language (Suhr et al., 2017), Visual Spatial Reasoning (Liu et al., 2023a) for spatial reasoning, LLaVA-Instruct-77K and SVIT, both generated by GPT-4 (OpenAI, 2023), for complex reasoning.

**Visual Dialog (VR)**  This task necessitates the model's capacity to chat naturally with humans about visual content. We include visual dialogue (Das et al., 2017), LLaVA-Instruct-58k and SVIT.

**Referential Dialogue (RD)**  This type of task is defined by Shikra (Chen et al., 2023b), which covers a wide range of vision-language positioning tasks for fine-grained image comprehension, including tasks: referring expression comprehension (REC) (Kazemzadeh et al., 2014; Mao et al., 2016), VQA grounding and grounding caption (GC) (Zhou et al., 2020), referring expression generation (REG) (Liu et al., 2017). Additionally, we also propose the Referential Complex Reasoning (RCR) task and Referential Detailed Description (RDD) task based on the SVIT datasets by employing ChatGPT to add objects' bounding boxes to their responses.

**Language-only tasks**  To maintain the chatting and text instruction-following capabilities of language models, we include tasks: Alpaca GPT-4 (Taori et al., 2023), Dolly 15k (Conover et al., 2023). For learning to reason, Flan-mini CoT (Ghosal et al., 2023), CodeX MathQA, CoQA (Reddy et al., 2019), Math (Li et al., 2023d) datasets are also incorporated. Note that we filter out some out-of-domain examples to avoid language models learning to refuse to answer image-related instructions.

## A.2  COMPARISON OF EXISTING VISUAL INSTRUCTION TUNING DATASETS

Currently, there are many novel models in visual instruction tuning, most of which are data-centric studies. Therefore, we summarize the datasets employed by the part of multimodal large language models (MLLMs) and compare them with those used by Octopus, shown in Table 7. Notably, the training process of many MLLMs consists of two stages (indicated by [†]), and we generally consider that the first stage is employed for aligning their visual and textual features, while the second stage is truly for visual instruction tuning. Therefore, we only consider the training data from the second stage for these models.

**LLaVA**  utilizes the COCO dataset, leveraging its caption and object bounding box information, to prompt GPT-4 in generating 158k high-quality visual instruction data. These instructions are categorized into three types: detailed descriptions, conversations, and complex reasoning.

**MiniGPT-4**  employs ChatGPT to paraphrase 5,000 image descriptions generated by raw MiniGPT-4 (trained in the first stage), and manually selected 3,500 high-quality image caption data from them.

**Otter**  proposes a dataset named MIMIC-IT, consisting of 47 different tasks, to enable the model to follow user instructions while preserving the in-context learning capability of OpenFlamingo. Each sample within the dataset contains few-shot templates for in-context instruction tuning. Additionally, MIMIC-IT serves as a multilingual visual instruction dataset.

**MultiModal-GPT** integrates datasets from LLaVA and MiniGPT-4, along with several traditional vision-language tasks, including A-OKVQA, COCO Caption, and OCR VQA. To maintain the model's ability to chat with humans, it also introduces language-only tasks, including Dolly 15k and the GPT-4 alpaca dataset.

**InstructBLIP** also employs the LLaVA dataset, but it's more focused on traditional vision-language tasks, aiming to improve its generalization performance to unseen tasks by increasing task diversity. In total, this work collects 26 vision-language tasks, with half of them used for training and the other half for evaluation.

**M³IT** assembles 40 traditional vision-language datasets, and it also belongs to a multilingual visual instruction dataset, spanning over 80 languages. Despite encompassing such a multitude of datasets, each one only contains a single task, thereby constraining its task diversity.

**Clever Flamingo** collects 37 vision-language tasks and utilizes the Polite Flamingo model (trained in the first stage) to rewrite the answers of these tasks into human-preferred responses, resulting in a high-quality PF-1M dataset that assists the model in reducing multimodal tax.

**Shikra** focuses on referential dialogue. It utilizes GPT-4 to generate 7,800 high-quality location-related question pairs which are considered as the Grounding Chain-of-Thought (GCoT) data based on the Flickr30K dataset. Additionally, it also incorporates 50% of the LLaVA dataset. As a result, there are a total of four types of tasks included.

**Emu** is trained to align with users' intentions and generalize to unseen tasks through visual instruction tuning, which gathers tasks from three distinct modalities: text, image, and video. For language-only tasks, the ShareGPT and Alpaca datasets are utilized. The LLaVA dataset is employed to align with image instructions, and the VideoChat dataset is introduced to assist the model in understanding video instructions.

Table 8: Detailed information of visual instruction tuning dataset we collected.

| Dataset | Task Name | #samples | | Source |
|---|---|---|---|---|
| | | Train | Valid | |
| MSCOCO (Lin et al., 2014) | Image Caption | 2,000 | 200 | MSCOCO |
| | Question-Image Matching | 2,000 | 200 | MultiInstruct |
| | Image-Text Selection | 2,000 | 200 | MultiInstruct |
| | Image-Text Matching | 2,000 | 200 | MultiInstruct |
| | Object Grounding | 2,000 | 200 | MultiInstruct |
| | Object-Region Matching | 2,000 | 200 | MultiInstruct |
| | Object-Region Selection | 2,000 | 200 | MultiInstruct |
| | Object Matching | 2,000 | 200 | MultiInstruct |
| | Missing Object Selection | 2,000 | 200 | MultiInstruct |
| IQA (Duanmu et al., 2021) | Image Quality | 2,000 | 200 | IQA |
| COCO-Text (Veit et al., 2016) | Text Type Classification | 2,000 | 200 | COCO-Text |
| | Text Legibility | 2,000 | 200 | COCO-Text |
| | Region-Text Matching | 2,000 | 200 | COCO-Text |
| | Text Localization | 2,000 | 200 | COCO-Text |
| Visual Genome (Krishna et al., 2017) | Grounded Captioning | 2,000 | 200 | Visual Genome |
| | Visual Grounding | 2,000 | 200 | Visual Genome |
| | Most-Overlapping Region Selection | 2,000 | 200 | MultiInstruct |
| | Non-Overlapping Region Selection | 2,000 | 200 | MultiInstruct |
| | Least-Overlapping Region Selection | 2,000 | 200 | MultiInstruct |
| | Overlapping Region Selection | 2,000 | 200 | MultiInstruct |
| | Region Overlapping Detection | 2,000 | 200 | MultiInstruct |
| | Region-Caption Matching | 2,000 | 200 | MultiInstruct |
| | Grounded Caption Selection | 2,000 | 200 | MultiInstruct |
| | Visual Grounding Selection | 2,000 | 200 | MultiInstruct |
| RefCOCO (Kazemzadeh et al., 2014) | Referring Expression Selection | 6,000 | 200 | MultiInstruct |
| RefCOCO+ | Referring Expression Generation | 6,000 | 200 | RefCOCO |
| RefCOCOg | Referring Expression Comprehension | 6,000 | 200 | RefCOCO |
| Visual7w (Zhu et al., 2016) | Grounded VQA | 2,000 | 200 | Visual7w |
| GQA (Hudson & Manning, 2019) | Visual Reasoning | 2,000 | 200 | GQA |
| OCR VQA (Mishra et al., 2019) | Reading Comprehension VQA | 2,000 | 200 | OCR VQA |
| OK-VQA (Marino et al., 2019) | Knowledge Grounded VQA | 2,000 | 200 | OK-VQA |
| VQAv2 (Goyal et al., 2017) | Open-Domain VQA | 2,000 | 200 | VQAv2 |
| TDIUC (Kafle & Kanan, 2017) | Sport Recognition | 2,000 | 200 | TDIUC |
| | Scene Recognition | 2,000 | 200 | TDIUC |
| | Color Attribute | 2,000 | 200 | TDIUC |
| | Other Attribute | 2,000 | 200 | TDIUC |
| | Activity Recognition | 2,000 | 200 | TDIUC |
| | Position Reasoning | 2,000 | 200 | TDIUC |
| | Object Recognition | 2,000 | 200 | TDIUC |
| | Absurd | 2,000 | 200 | TDIUC |
| | Utility and Affordance | 291 | 143 | TDIUC |
| | Object Presence | 2,000 | 200 | TDIUC |
| | Counting | 2,000 | 200 | TDIUC |
| | Sentiment | 1,242 | 200 | TDIUC |
| LLaVA-Instruct (Liu et al., 2023b) | Detailed Description | 3,500 | - | LLaVA |
| SVIT-Instruct (Zhao et al., 2023a) | Complex Reasoning | 3,500 | - | LLaVA |
| | Conversation | 7,000 | - | LLaVA |
| Otter-LA-I (Li et al., 2023a) | LLaVA-Interleaved | 7,000 | - | Otter |
| Shikra-RD (Chen et al., 2023b) | Referential Dialogue | 7,800 | - | Shikra |
| GQA-CoT | Referential Dialogue GCoT | 5,000 | - | Shikra |
| Octopus | Referential Detailed Description | 5,000 | - | Octopus |
| (Generated data from SVIT) | Referential Complex Reasoning | 3,971 | - | Octopus |
| Alpaca (Taori et al., 2023) | Human Preferred Text Instruction | 4,000 | - | Alpaca |
| Dolly (Conover et al., 2023) | | | | |
| MathQA (Amini et al., 2019) | CodeX Math QA | 2,000 | - | - |
| Flan-Mini (Ghosal et al., 2023) | Flan-CoT | 2,000 | - | Flan |
| CoQA (Reddy et al., 2019) | Conversational QA | 5,000 | - | CoQA |
| Math (Li et al., 2023d) | Math | 2,000 | - | Camel-AI |

### A.3 PROMPT TEMPLATES STATISTICS

**Data Filtering Strategy.** Using natural language to describe position information would significantly occupy the prompt length. Considering the most extreme scenario, we need to incorporate in-context exemplars responses, leading to excessive prompt length for certain examples. Therefore, we filter out these examples. Additionally, for language-only tasks, we exclude examples containing words or phrases like "image" and "I'm sorry," aiming to prevent our model from hallucination and learning to refuse instructions related to images. Finally, this filtering process reduces our actual training examples from 148,304 to 121,184.

**In-context Exemplars.** We utilize the approach described in Section 2.4 to select in-context exemplars for examples in vision-language tasks. In practice, we set $k$ to 3, meaning that for each example, we select 6 in-context exemplars (3 image exemplars and 3 text exemplars) based on cosine similarity. Each exemplar cosine similarity must surpass a certain threshold to be considered, with thresholds of 0.7 and 0.8 for image and text exemplars, respectively. For some tasks, such as NLP tasks, we randomly select in-context exemplars that are not counted in the final few-shot template quantity.

Table 9: Exemplars with different types of prompts, including CoT and few-shot templates.

|  | #Samples | Pct. (%) |
| --- | --- | --- |
| Total samples | 155,304 | 100.00% |
| Filtered samples | 148,304 | 95.49% |
| Few-shot templates | 51,513 | 33.17% |
| Few-shot templates (image) | 24,945 | 16.06% |
| Few-shot templates (text) | 26,568 | 17.11% |

| Dataset | Task | Split | Samples | Metric |
| --- | --- | --- | --- | --- |
| Flickr30K | Scene description | Test | 1000 | CIDEr(↑) |
| OK-VQA | Knowledge Grounded VQA | Val | 5046 | VQA acc.(↑) |
| TextVQA | Reading Comprehension VQA | Val | 5000 | VQA acc.(↑) |
| ScienceQA | Visual Reasoning QA | Test | 2017 | Accuracy(↑) |
| HatefulMemes | Image Classification | Test | 1000 | ROC-AUC(↑) |
| VizWiz | Scene Perception | Val | 4319 | VQA acc.(↑) |
| ISEKAI | Link-Context Learning | Test | 1256 | Accuracy(↑) |

Table 10: Description of dataset in zero- and few-shot evaluation benchmarks.

## B EVALUATION METRICS

We provide evaluation metrics as Table 10. For the Flickr30K dataset, we report the CIDEr score to assess model performance using the coco-caption evaluation code [3]. Regarding ScienceQA, following Dai et al. (2023), we remove samples without images and employ standard evaluation code [4] to calculate top-1 accuracy (%). It's important to note that for all tasks, we refrain from adding vocabulary constraints and rankings approaches.

## C MODEL

**Architecture.** Our model is based on OpenFlamingo architecture, named `UMIT`. The model consists of a vision encoder from CLIP (Radford et al., 2021), a perceiver resampler to receive the visual

---

[3]https://github.com/tylin/coco-caption
[4]https://github.com/lupantech/ScienceQA

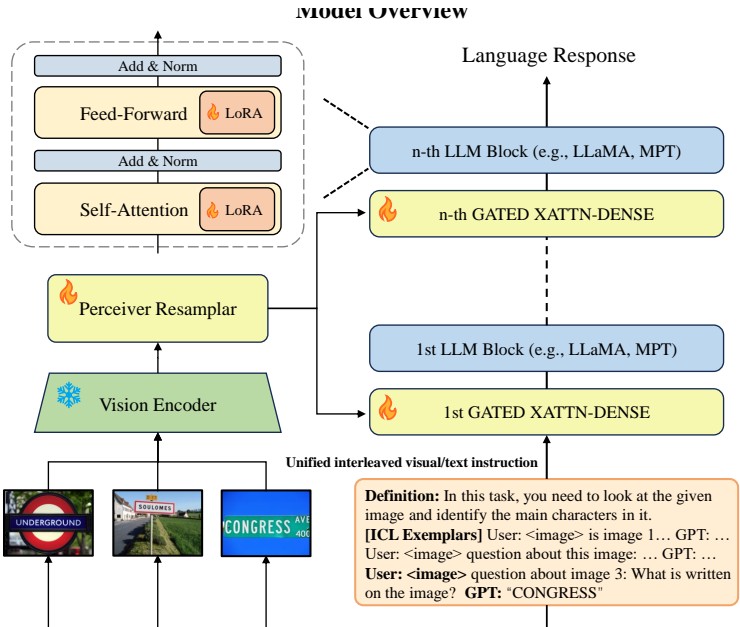

Figure 4: Our model overview.

features from the vision encoder, and a text encoder from large language models (e.g., LLaMA (Touvron et al., 2023), MPT (Team, 2023)) equipped with gated cross-attention layers for image-text interactions. To preserve the pretraining knowledge and reduce computing consumption, We freeze both the encoders and only finetune the perceiver resampler module, gated cross-attention layers, and LoRA (Hu et al., 2021) module added to the text encoder, as shown in Figure 4.

**Training Details.**   `UMIT` is trained in two stages. The first stage aims to facilitate the model's capability to comprehend numerical positional coordinates in natural language. We train `UMIT` for 2 epochs (approximately 10k steps), using several datasets encompassing RefCOCO for REC and REG tasks, Visual Genome for GC tasks, and our newly introduced RDD and RCR tasks (Appendix A.1). In the second stage, we train `UMIT` on our proposed dataset with only 155k samples but consisting of 55 tasks in `DEIQA` format ($\mathcal{F}_3$). It takes around 30h for stage one training and 6h for stage two.

**Training Strategy for In-Context Learning.**   After retrieval, every example from most tasks has $2k$ in-context exemplars to construct the unified multimodal instruction format. However, adding in-context exemplars can significantly increase the length of the prompt, leading to sparsity in the loss and lower batch diversity. Therefore, we follow Iyer et al. (2022) to in-context instruction-finetune our model with suffix loss rather than the original MetaICL loss, as illustrated in Table 11. The difference between the two is that the former only calculates the loss of the last target answer, while the latter needs to calculate the loss from the first exemplars' answer and the remaining part, which mitigates the problem of loss sparsity.

# D   TASK DEFINITION

Task definitions generated by GPT-3.5 for MME Benchmark, as shown in Table 10.

Task definitions generated by GPT-3.5 for SEED-Bench, as shown in Table 13.

```
Definition: [Task Definition (task-level instruction)]
# in-context exemplars
User: <image>question about image 1:[question]
GPT: <answer> [answer] <endofchunk>
...
User: <image> question about image 3:[question]
GPT: <answer> [answer] <endofchunk>
# current instance
User: <image> question about this image:[question]
GPT: <answer> [answer] <endofchunk>
```

Table 11: `<image>` and `<endofchunk>` tokens are originally from the OpenFlamingo training paradigm, and we follow Li et al. (2023b) to include a new token `<answer>` for intercepting the target answer of the model output more easily, which can also assist in adjusting in-context learning loss mentioned in Appendix A.3. Note that only green sequence/tokens are used to compute the loss and we train our model using a cross-entropy loss and .

- **Existence**: In this task, you will receive an image and a question about the existence of an object in the image.

- **Count**: In this task, you will need to compare the number of objects in a given image with the sentence given, and determine whether the number is correct or not.

- **Position**: In this task, you are given a sentence and a image. You need to judge whether the sentence correctly describes the positional relationships between objects in the diagram.

- **Color**: In this task, you need to look at the given image and answer whether the color described in the question matches the content of the image.

- **Posters**: In this task, you need to determine if the description of the given question matches the information of a poster image.

- **Celebrity**: In this task, you will receive an image and a question about the name of the celebrity selected by a red bounding box in this image.

- **Scene**: In this task, you need to look at the image and determine whether the description of the image in the given question is correct or not.

- **Landmark**: In this task, you need to evaluate whether the given question is correctly describing the name of the landmark in the image.

- **Artwork**: In this task, given a picture of an artwork and a question about it, you need to answer the question based on the content of the picture and your own general knowledge.

- **OCR**: In this task, you need to evaluate whether the characters mentioned in a sentence are the same as the ones in a given image.

Table 12: The list of task definitions for MME Benchmark.

- **Scene Understanding**: In this task, you need to analyze the scene in the provided image and answer a given question.

- **Instance Identity**: In this task, you will receive an image and please answer a question about the existence of some objects in the image.

- **Instance Attributes**: In this task, you need to answer questions about the attribute of an object in the following images.

- **Instance Location**: In this task, you are required to understand the location of objects in the image and answer a question about this location information in the image.

- **Instance Counting**: In this task, you need to count the number of objects in this image and answer the corresponding question.

- **Spatial Relations**: In this task, you need to answer a question by understanding the spatial relations between objects in a given image.

- **Instance Interaction**: In this task, you need to find two objects in a given image and then reason out their relation by looking at the image to answer a question.

- **Visual Reasoning**: In this task, you need to answer the given questions by referring to the content of the image and using commonsense reasoning.

- **Text Recognition**: In this task, you are given an image with some texts on it and you need to locate the texts in the image and answer a corresponding question related to them.

Table 13: The list of task definitions for SEED-Bench.

