# OpenReview forum: "Understanding Multimodal Instruction Format for In-context Learning"
_ICLR.cc/2024/Conference — Submitted to ICLR 2024_

### Official Review · Reviewer_y2C2 · 2023-10-30

**Soundness:** 2 fair
**Presentation:** 2 fair
**Contribution:** 2 fair
**Rating:** 5
**Confidence:** 5

**Summary:**

This work explores the impact of different instruction formats on the in-context learning ability of MLLM. The authors propose a UMIT method, which introduces task definition and transfers the multimodal instructions of diverse tasks in a unified style. A retrieval-based approach is also proposed to enhance exemplars sampling.

**Strengths:**

1. This paper is the first to investigate the impact of different instruction formats on the in-context learning ability of MLLM.
2. The proposed UMIT outperforms OpenFlamingo and Otter in terms of performance on the MME Benchmark and SEED-Bench.

**Weaknesses:**

1. This work mainly focuses on instruction formats, lacks innovation, and fixed instruction formats are difficult to generalize to open-domain tasks.
2. When testing, does UMIT require using ChatGPT to obtain the task definition for each new sample? This can result in significant inference costs for the unseen task.
3. The description of the details of UMIT, especially the use of symbols, is somewhat confusing. For example, what does the text encoder encode in exemplar sampling? And where does X_{instruct}^{i} come from in section 2.5?
4. The experimental results on SEED-Bench show a significant improvement when training OpenFlamingo and Otter on the data collected by the authors. In contrast, the gains from changing the format seem less pronounced. This raises a question of whether the role of data diversity is much greater than the task definition proposed by the author.
5. Some experimental results on VizWiz contradict the author's conclusions and should be analyzed in more detail. For example, in Table 4, "DEIQA (F3) mixed" performs worse than "DEIQA (F3) random". And in Table 5, "vqa+diff" performs worse than "vqa+same".

**Questions:**

1. In Otter / MMICL, what is the difference between "instance-level instruction" and "question"?
2. Do other ICL methods harm zero-shot performance?

---

> ### Author Response · Authors · 2023-11-22
> **Author Response for Reviewer y2C2 (Part 1)**
>
> Dear Reviewer y2C2:
>
> Thank you for your comments. We provide discussions and explanations about your concerns as follows.
>
> **Q1: This work mainly focuses on instruction formats, lacks innovation, and fixed instruction formats are difficult to generalize to open-domain tasks.**
>
> **A1:** We acknowledge your concerns regarding the generalization of our method to open-domain tasks. Here, we categorize the open-domain tasks you mentioned into two types. The first type belongs to traditional vision-language tasks, such as open-domain Visual Question Answering (VQA) tasks like GQA [16], VQAv2 [17], etc. Our proposed UMIT can generalize to these tasks, which has been demonstrated by our results shown in Table 6.
> The second type is real-world tasks. We must acknowledge that, since real-world instructions are diverse and challenging to describe in terms of task definition, our framework currently has limitations when extending to such tasks. However, we contend that, at present, existing MLLMs other than GPT-4V lack in-context learning abilities for real-world tasks. Simultaneously, there is a lack of a reasonable benchmark to effectively evaluate the in-context learning ability of MLLMs in real-world tasks and provide quantifiable results. Therefore, this should be considered a common limitation in the current field of multimodal large language models, rather than being exclusive to our work. A detailed discussion can be found in our comments to reviewer 4eXe in **A2**.
>
>
> **Q2: When testing, does UMIT require using ChatGPT to obtain the task definition for each new sample? This can result in significant inference costs for the unseen task.**
>
> **A2:** No, during the testing stage, UMIT only requires using ChatGPT to obtain the task definition for each task rather than each sample. As demonstrated in Table 12 and Table 13, we showcase the task definitions we obtained for tasks from the MME benchmark and SEED Bench using ChatGPT before the inference stage. This process is handled before inference, so it does not add to the inference costs.
>
> **Q3: The description of the details of UMIT, especially the use of symbols, is somewhat confusing. For example, what does the text encoder encode in exemplar sampling? And where does X_{instruct}^{i} come from in section 2.5?**
>
> **A3:** Sorry for this confusion. We will revise the paper to clarify them.
> + we use text embeddings of both question X_q and answer X_a to retrieve in-context exemplars in training stage. This ensures that we can still retrieve similar in-context examples in tasks with specific questions other than VQA tasks. And for all evaluations, we randomly select in-context exemplars in testing stage.
> + All X_{instruct}^{i} are derived from annotations in the original datasets we collected. In our response **A1** to **reviewer 4eXe**, we provided a detailed list of three instances from three distinct multimodal instruction datasets. We hope that reviewing these instances will help clarify this confusion.
>
>
> **Q4: The experimental results on SEED-Bench show a significant improvement when training OpenFlamingo and Otter on the data collected by the authors. In contrast, the gains from changing the format seem less pronounced. This raises a question of whether the role of data diversity is much greater than the task definition proposed by the author.**
>
> **A4:** We highly agree with your perspective. At least for the present, the improvement brought about by increasing task diversity on traditional vision-language metrics is indeed irreplaceable. This is also confirmed by LLaVA 1.5 [18], and in the NLP community, it has also been proven, as seen in works like Flan-2022 [19], opt-iml [20], etc. The purpose of introducing "task definition" is also to effectively merge existing multimodal instruction datasets using a unified format. This is done to enhance task diversity while, as much as possible, preserving the ability for in-context learning without compromising zero-shot performance.

---

> > ### Author Response · Authors · 2023-11-22
> > **Author Response for Reviewer y2C2 (Part 2)**
> >
> > **Q5: Some experimental results on VizWiz contradict the author's conclusions and should be analyzed in more detail. For example, in Table 4, "DEIQA (F3) mixed" performs worse than "DEIQA (F3) random".**
> >
> > **A5:** We apologize for the insufficient analysis, and we will revise the paper to provide a clearer analysis of these results. The performance drop from "random" to "mixed" might be attributed to the random selection of in-context examples during the testing stage, which could introduce a certain level of bias. Additionally, on the other two datasets, HatefulMemes and ISEKAI, the performance of "DEIQA (F3) mixed" and "DEIQA (F3) random" is comparable. Therefore, the bias introduced by in-context examples might result in a slightly inferior performance for "mixed" compared to "random." To investigate this phenomenon, we conducted three random seed selections for in-context examples and obtained the following results:
> >
> > | **Method**           | **0** | **2** | **4** | **8** |
> > |----------------------|-------|-------|-------|-------|
> > | DEIQA + Random (avg) | **27.15** ±0.14 | 29.53 ±1.06 | 30.66 ±0.86 | 31.21 ±0.67 |
> > | DEIQA + Mixed (avg)  | 27.03 ±0.06 | **29.66** ±0.98 | **31.02** ±0.96 | **31.87** ±0.29
> >
> >
> >
> > **Q6: And in Table 5, "vqa+diff" performs worse than "vqa+same".**
> >
> > **A6:** We contend this result is within an acceptable range. This is because the VizWiz dataset belongs to the VQA task category: it just requires the model to answer image-related questions except questions related to blurry images. Therefore, the performance improvement from adding tasks outside the VQA is expected to be less significant than the improvement from increasing task diversity within the VQA. To further evaluate the performance improvement from increased task diversity, we selected three tasks beyond VQA for evaluation, including Visual Spatial Reasoning (only choose 1k examples) [21], ScienceQA, Flickr30K. The results are as follows:
> >
> > | Dataset   | vqa + same | vqa + diff  | all               |
> > |-----------|------------|-------------|-------------------|
> > | VSR (1k)       | 52.62      | 54.09       | **57.69**             |
> > | ScienceQA | 57.40      | 59.04       | **59.55**             |
> > | Flickr30K | 61.03      | 67.29       | **68.22**             |
> >
> > All the result are zero-shot performance. As evident, due to the incorporation of the COCO caption task type in "diff," while "same" consists only of VQA tasks, "VQA+diff" performs significantly better on the image caption task of Flickr30K compared to "VQA+same."
> >
> >
> >
> > **Q7: In Otter / MMICL, what is the difference between "instance-level instruction" and "question"?**
> >
> > **A7:**  In our response **A1** to **reviewer 4eXe**, we provided a detailed list of three instances from three distinct multimodal instruction datasets, including Otter dataset, MMICL dataset, MultiInstruct dataset. We hope that reviewing these instances will help clarify this confusion.
> >
> >
> > **Q8: Do other ICL methods harm zero-shot performance?**
> >
> > **A8:** Currently, works that focus on multimodal in-context learning during the instruction-tuning phase include Otter and MMICL. They both do not harm zero-shot performance; instead, it can be argued that they are more concerned with the outcomes of zero-shot performance, to some extent, neglecting few-shot performance. For instance, Otter only showcases its few-shot performance on the COCO caption task, and MMICL only presents the difference in 0-shot and 4-shot performance on 4traditional VL tasks. In contrast, our method demonstrates performance variations from 0-shot to 8-shots on the MME benchmark (10 tasks), SEED Bench (9 tasks), and three traditional visual-linguistic tasks.
> >
> > [16] Hudson, Drew A. and Christopher D. Manning. “GQA: A New Dataset for Real-World Visual Reasoning and Compositional Question Answering.” 2019 IEEE/CVF Conference on Computer Vision and Pattern Recognition (CVPR) (2019): 6693-6702.
> > [17] Goyal, Yash et al. “Making the V in VQA Matter: Elevating the Role of Image Understanding in Visual Question Answering.” International Journal of Computer Vision 127 (2016): 398 - 414.
> > [18] Liu, Haotian et al. “Improved Baselines with Visual Instruction Tuning.” ArXiv abs/2310.03744 (2023): n. pag.
> > [19] Longpre, S. et al. “The Flan Collection: Designing Data and Methods for Effective Instruction Tuning.” International Conference on Machine Learning (2023).
> > [20] Iyer, Srinivas et al. “OPT-IML: Scaling Language Model Instruction Meta Learning through the Lens of Generalization.” ArXiv abs/2212.12017 (2022): n. pag.
> > [21] Liu, Fangyu et al. “Visual Spatial Reasoning.” Transactions of the Association for Computational Linguistics 11 (2022): 635-651.

---

> > > ### Author Response · Authors · 2023-11-23
> > >
> > > Dear Reviewer y2C2,
> > >
> > > Thank you again for your invaluable insights and observations! We would be immensely grateful if you could share your satisfaction level regarding our response. We will be happy to address any remaining concerns.
> > >
> > > Sincerely,
> > >
> > > Paper8964 Authors

---

### Official Review · Reviewer_srth · 2023-10-30

**Soundness:** 2 fair
**Presentation:** 2 fair
**Contribution:** 2 fair
**Rating:** 5
**Confidence:** 4

**Summary:**

This paper aims to find a better vision-language instruction format for in-context learning. Based on the existing components of [examples, image, instruction, question, answer] in instruction format, the authors further introduce the task definition as the prefix of the instruction. During the test, the authors also explored different types of exemplar sampling methods. The models are compared with the previous OpenFlamingo and Otter models.

**Strengths:**

- The problem of enhancing the in-context learning capabilities of the vision-language model is important.
- The proposed method of adding a task definition is very simple.

**Weaknesses:**

- The contribution and novelty of this work is insufficient. Adding a carefully designed task definition with minor improvements (0.3 in Tab3) is not that significant in terms of technical contributions or scientific findings.
- Although the paper uses a mix of many existing datasets for training, the evaluation is limited to a few benchmarks.
- The writing is not that good. For example, the name Octopus is not introduced in the paper, which is confusing. And some typos need to be revised.
- The literature review in Sec2.1 is not correct, e.g., "Moreover, Otter (Li et al., 2023b) and MMICL (Zhao et al., 2023b) further introduced". Otter was before Qwen-VL.

**Questions:**

Please refer to the weaknesses.

---

> ### Author Response · Authors · 2023-11-22
> **Author Response for Reviewer srth**
>
> Dear Reviewer srth:
>
> Thank you for your comments. We provide discussions and explanations about your concerns as follows.
>
> **Q1: The contribution and novelty of this work is insufficient. Adding a carefully designed task definition with minor improvements (0.3 in Tab3) is not that significant in terms of technical contributions or scientific findings.**
>
> **A1:** Thank you for your careful examination of our table. In fact, the UMIT framework is designed to efficiently merge different multimodal instruction datasets with various tasks using a unified format, preserving the model's in-context learning ability on traditional vision-language tasks to the greatest extent possible without damaging the zero-shot performance of MLLMs.
>
> We contend that multimodal instruction datasets are currently in a nascent stage of development, with significant limitations in terms of task diversity. Currently, the multimodal instruction dataset with the highest number of tasks is vision-flan [13], boasting over 200 tasks. However, this is still considerably fewer than instruction datasets in the NLP community, such as Flan-2022 [14] (1836 tasks) and Ni-v2 [15] (1616 tasks). Therefore, to rapidly increase task diversity, we can merge existing multimodal instruction datasets. Designing a unified format is what we consider the foremost requirement for merging these datasets, and this is the significance behind the birth of the UMIT framework.
>
> Regarding Table 3, our primary focus has been to investigate the impact of the Unified Multimodal Instruction Format (UMIT) on multimodal in-context learning. In Table 3, our format demonstrates a performance improvement of 1.2 through in-context learning, while the otter-style format yields a more modest improvement of 0.5. Although the OpenFlamingo-style format exhibits a stronger in-context learning capability, with an improvement of 2.5, this is due to its lack of instructions. Consequently, it minimizes the gap between fine-tuning and pre-training data format, preserving in-context learning to the greatest extent. However, the consequence is a significant impact on its zero-shot performance. Therefore, we consider our format to be the optimal compromise.
>
> Moreover, given that SEED Bench comprises 15,425 examples, we argue that both the 0.7 improvement in few-shot performance compared to the Otter-style format and the remarkable 3.2 improvement in zero-shot performance compared to the OpenFlamingo-style format are highly significant and noteworthy.
>
>
> **Q2: Although the paper uses a mix of many existing datasets for training, the evaluation is limited to a few benchmarks.**
>
>
> **A2:** Thank you very much for acknowledging the diversity of our training dataset. However, for evaluation, we conducted tests on the MME Benchmark with 14 tasks and the SEED Bench with 12 tasks, as shown in Tables 2 and 3. Additionally, we performed zero- and few-shot tests on seven traditional visual-linguistic tasks, including VizWiz, hatefulmeme, ISEKAI, Flickr30K, ScienceQA-Image, OK-VQA, and Text-VQA, as shown in Tables 4 and 6. If you feel that our experiments are not comprehensive, we sincerely hope you could recommend additional benchmarks or datasets, and we will make every effort to conduct thorough testing on these datasets.
>
>
> **Q3: The writing is not that good. For example, the name Octopus is not introduced in the paper, which is confusing. And some typos need to be revised.**
>
> **A3:** Sorry for the confusion. We will revise the article to change "Octopus" to "UMIT".
>
> **Q4: The literature review in Sec2.1 is not correct, e.g., "Moreover, Otter (Li et al., 2023b) and MMICL (Zhao et al., 2023b) further introduced". Otter was before Qwen-VL.**
>
> **A4:** We apologize for this error. It occurred because we wrote in the order of the increasing complexity of their instruction format components. In the Qwen-VL format, there is a lack of instructions, while Otter includes instructions, so we placed Qwen-VL first. We will revise the article to correct this error in Section 2.1. Part of the revised text is as follows:
> > Unlike the format utilized by Qwen-VL, Otter and MMICL incorporated the instance-level instruction component into the prompt format design for in-context instruction tuning.
>
> [13] [Xu, Trevor Ashby et al. “Vision-Flan: Scaling Visual Instruction Tuning.”](https://vision-flan.github.io/)
> [14] Longpre, S. et al. “The Flan Collection: Designing Data and Methods for Effective Instruction Tuning.” International Conference on Machine Learning (2023).
> [15] Wang, Yizhong et al. “Super-NaturalInstructions: Generalization via Declarative Instructions on 1600+ NLP Tasks.” Conference on Empirical Methods in Natural Language Processing (2022).

---

> > ### Author Response · Authors · 2023-11-23
> >
> > Dear Reviewer srth,
> >
> > Thank you again for your invaluable insights and observations! We would be immensely grateful if you could share your satisfaction level regarding our response. We will be happy to address any remaining concerns.
> >
> > Sincerely,
> >
> > Paper8964 Authors

---

### Official Review · Reviewer_Sdyx · 2023-11-02

**Soundness:** 2 fair
**Presentation:** 2 fair
**Contribution:** 3 good
**Rating:** 6
**Confidence:** 4

**Summary:**

This paper proposes Unified Multimodal Instruction Tuning (UMIT), a framework to suggest how to construct a text-image interleaved instruction dataset by merging diverse visual instruction datasets in a unified multimodal instruction format. The experiments are based on OpenFlamingo. This paper also studies the impact of different components in multimodal instruction formats.

**Strengths:**

The proposed approach seems reasonable for certain tasks; studying how the format of the multimodal instruction will affect the in-context learning performance is also an interesting topic. The experiments show positive results on the benchmarked datasets and tasks.

**Weaknesses:**

One concern the reviewer has is that: the strategy of using uniform instruction styles and defining tasks clearly helps with tasks we already know well. However, this method assumes we can list and describe every possible task type, which might not be practical. The real world is full of unexpected and varied tasks, and a model that's too focused on a set of specific tasks might struggle to adapt to new or different ones. This over-specialization could limit the model's usefulness in a wider range of real-world situations.
Also, if the reviewer understands it correctly, the task definition for each instruction-exemplar pair is manually crafted, which might introduce empirical errors/bias, and might not be scalable enough.

**Questions:**

How will the quality of the manually crafted task definition affect the overall performance? Have the authors tried efforts to automate this process to make it maybe more scalable?

---

> ### Author Response · Authors · 2023-11-22
> **Author Response for Reviewer Sdyx (Part 1)**
>
> Dear Reviewer Sdyx:
>
> Thank you for your comments. We provide discussions and explanations about your concerns as follows.
>
>
> **Q1: If the reviewer understands it correctly, the task definition for each instruction-exemplar pair is manually crafted, which might introduce empirical errors/bias, and might not be scalable enough.**
>
> **A1:** Sorry for the confusion. We acknowledge that the description in Section 2.3 may have been a bit unclear. In reality, we did not manually create a "task definition" for each instruction-exemplar pair. As illustrated in Table 7, our dataset comprises a total of 55 tasks from various multimodal instruction datasets, and for each of these tasks, there exists a specific "task definition." These task definitions may originate from annotations in the original dataset, such as MultiInstruct dataset [10]. If the original data lacks annotations, we manually write the "task definition", such as OCR-VQA [11]. In essence, we manually craft "task definition" for just around 30 tasks. Additionally, to reduce bias caused by style inconsistency in various task definitions, we used GPT-3.5 to transform these 55 "task definitions" into a consistent style: "In this task, you need to...". Therefore, the "task definition" for each instruction-exemplar pair is derived from the task to which they belong.
>
>
> **Q2: This method assumes we can list and describe every possible task type, which might not be practical. The real world is full of unexpected and varied tasks, and a model that's too focused on a set of specific tasks might struggle to adapt to new or different ones. This over-specialization could limit the model's usefulness in a wider range of real-world situations.**
>
> **A2:** We highly agree with your perspective on the important role of in-context learning in the real world tasks. Since real-world instructions are quite diverse and hard to describe their task definition, we instead focus on instructions that describe traditional vision-language tasks to rigorously study the effects of different components of multimodal instruction format on in-context learning ability. Furthermore, unlike LLMs, current MLLMs, with the exception of GPT-4V, have not demonstrated a robust in-context learning ability in complex real-world tasks. A detailed discussion can be found in our comments to reviewer 4eXe in **A2**.
> Furthermore, our evaluations are all conducted on tasks that the model has not seen during training stage, such as the MME benchmark in Table 2 and the SEED Bench in Table 3. The results indicate that our model's generalization capabilities on few-shot performance remains significant. Therefore, once MLLMs with demonstrated in-context learning ability in the real world are open-sourced and there is a benchmark for MLLMs' in-context learning ability in the real world, we believe that the insights derived from our study could potentially be applied to real-world tasks and yield quantifiable results.

---

> > ### Author Response · Authors · 2023-11-22
> > **Author Response for Reviewer Sdyx (Part 2)**
> >
> > **Q3: How will the quality of the manually crafted task definition affect the overall performance?**
> >
> > **A3:** As described in A1, we only manually crafted approximately 30 task definitions and utilized GPT-3.5 to transfer them all into a unified style. Thus, during the training stage, we ensure that the "task definitions" for 55 tasks in our dataset are of high quality.
> >
> > To assess the impact of task definition quality on overall performance, taking into account cost and efficiency, we only adjust the task definition quality during the testing stage for tasks in the MME benchmark. We then test their performance on the MME Benchmark, and here are the results:
> >
> > | **Quality of Task Definition** | **existence** | **count** | **position** | **color** | **posters** | **celebrity** | **scene** | **landmark** | **artwork** | **OCR** | **Avg** |
> > |--------------------------------|---------------|-----------|--------------|-----------|-------------|---------------|-----------|--------------|-------------|---------|---------|
> > | Random                         | 165.00        | 70.00     | 66.67        | 95.00     | 111.56      | 105.88        | 130.75    | 120.50       | 98.75       | 67.50   | 1031.61 |
> > | Raw                            | 165.00        | 66.67     | 66.67        | 111.67    | 112.59      | 113.24        | 137.25    | 124.25       | 100.25      | 60.00   | 1057.57 |
> > | UMIT                           | 180.00        | 53.33     | 48.33        | 103.33    | 138.10      | 129.41        | 157.25    | 126.00       | 95.00       | 65.00   | 1095.76 |
> >
> >
> > Where "UMIT" represents our use task definitions shown in Table 12. "Raw" indicates that we set the task definitions for all 10 tasks on the MME benchmark as: "In this task, you need to look at the image and answer the corresponding question." "Random" signifies that we randomly assigned the 10 task definitions shown in Table 12 to the 10 tasks on the MME benchmark. The results indicate that, during the testing stage, task definition quality does indeed have an impact on overall performance. However, this is observed under relatively extreme conditions, leading us to believe that task definitions, after manual inspection, exhibit robustness.
> >
> >
> > **Q4: Have the authors tried efforts to automate this process to make it maybe more scalable?**
> >
> > **A4:** In fact, the process of generating task definitions is automated, assuming the exclusion of diverse real-world tasks. For existing traditional VL tasks in multimodal datasets, each task invariably comes with its own description to assist users in understanding and utilizing the dataset. For instance, the SEED Bench [12] encompasses 12 tasks, and in its paper, there is a detailed description for each task. Therefore, all we need to do is take this description and input it along with our manually crafted task definition as context into GPT-3.5, as shown in Figure 1 (right). This allows us to directly obtain task definitions in a similar style without the need for manual crafting.
> >
> > [12] Li, Bohao et al. “SEED-Bench: Benchmarking Multimodal LLMs with Generative Comprehension.” ArXiv abs/2307.16125 (2023): n. pag.

---

> > > ### Comment · Reviewer_Sdyx · 2023-11-22
> > >
> > > Thank you for the clarifications. I've read other reviews and rebuttals.

---

### Official Review · Reviewer_4eXe · 2023-11-03

**Soundness:** 2 fair
**Presentation:** 3 good
**Contribution:** 2 fair
**Rating:** 5
**Confidence:** 4

**Summary:**

The paper proposes a new format for multimodal instruction tuning that includes a task definition prompt into the input context for multimodal models that can do in-context learning.

**Strengths:**

- The contribution of retrieval to select the best exemplars for in-context learning is interesting, and the results in Table 4 show the benefit of retrieving relevant in-context examples.

- The main contribution of "task definition" does not seem to be particularly novel, but the results in Tables 3 and 4 do seem to indicate the benefit of using task descriptions.

**Weaknesses:**

- The main contribution here seems to be the new "task definition" component of the prompt that precedes the in-context examples, which does not seem particularly novel, and looking at the examples in Figure 1, I don't really see what information they provide that is not provided in the instance instruction itself -- the task definition just seems like a more verbose form of the instance-level instruction.

- The retrieval augmentation in Section 2.4 requires us to have a reasonably large database of training examples to choose from, which may enhance task performance on benchmarks but defeats the purpose of in-context learning in the real world.

One ablation experiment I would suggest is to use the task-definition prompt without the instance-level instruction (DEQA).

**Questions:**

- In equation 5, is the text embedding E_text computed using the question X_q, answer X_a or both? this is not made clear

---

> ### Author Response · Authors · 2023-11-22
> **Author Response for Reviewer 4eXe (Part 1)**
>
> Dear Reviewer 4eXe:
>
> Thank you for your comments. We provide discussions and explanations about your concerns as follows.
>
> **Q1: The difference between the extra information provided about "task definition" and "instance-level instruction".**
>
> **A1**: We understand your concerns about whether adding the "task definition" component can enhance information beyond that provided by "instance-level instruction.". Firstly, we would like to illustrate the differences between the three components: "task definition," "instance-level instruction," and "question," by presenting an instance from one of the three different multimodal instruction datasets:
> 1. **Otter-LA dataset**. This dataset lacks the "task definition" component. And the majority of this dataset is derived from real-world tasks, so we consider the "instruction" and the "question" as the same component.
> ```
> ### Otter-LA dataset ###
>
> [component 1] Task Definition: None
>
> [component 2] In-context Exemplars:
> Instruction 1: None
> Question1: What skill set might someone need to perform such a frisbee trick?
> Answer1: To perform the frisbee trick shown in the image, where the man is passing a frisbee between or underneath his legs, a person would need a combination of skills...
> Instruction 2: None
> Question2: ...
> Answer2: ...
> ...
>
> [component 3] Instruction: None (you can consider the "question" as an "instance-level instruction".)
> [component 4] Question: What skills or techniques might the man need to successfully play with the frisbee?
> [component 5] Answer: To successfully play with the frisbee, the man needs various skills and techniques such as hand-eye coordination, spatial awareness, and understanding the aerodynamics...
> ```
>
> 2. **MMICL dataset**: This dataset also lacks the "task definition" component. However, the majority of its examples come from traditional vision-language tasks, such as OK-VQA, COCO-caption, etc. Therefore, we can distinguish its "instruction" component from the "question" component.
> ```
> ### MMICL dataset (OK-VQA) ###
>
> [component 1] Task Definition: None
>
> [component 2] In-context Exemplars:
> Instruction 1: Make sure your answers are based on the information presented in the image 0: <image0>图.
> Question 1: What is the purpose of the red decorations in this photo?
> Answer 1: lighting
> Instruction 2: ...
> Question 2: ...
> Answer 2: ...
>
> [component 3] Instruction: Carefully examine image 2 labeled <image2>图 before answering the question.
> [component 4] Question: Name the type of plant this is?
> [component 5] Answer: vine
> ```
>
> 3. **MultiInstruct dataset**: This dataset encompasses 62 traditional vision-language tasks, with 5 task-level instructions manually crafted for each task, referred to as "task definition." Therefore, it does not require additional "instance-level instruction." However, it lacks the "in-context exemplars" component.
> ```
> ### MultiInstruct dataset (OK-VQA) ###
>
> [component 1] Task Definition: In this task, you will be asked a question about image. However, in order to answer this question, you need knoweldge outside of this image.
>
> [component 2] In-context Exemplars: None
>
> [component 3] Instruction: None (you can consider the "task definition" as an "task-level instruction".)
> [component 4] Question: Name the type of plant this is?
> [component 5] Answer: vine
> ```
>
> Therefore, in our definition, the "task definition" component contains background information about the task, while the "instruction" component only guides the model to use the image content to answer relevant questions, without containing specific task information. Thus, with the addition of "task definition," we can even consider "instance-level instruction" and "question" as the same component.
>
> In fact, the UMIT framework is designed to efficiently merge different multimodal instruction datasets with various tasks using a unified format, preserving the model's in-context learning ability on traditional vision-language tasks to the greatest extent possible without damaging the zero-shot performance of MLLMs.
>
> We contend that multimodal instruction datasets are currently in a nascent stage of development, with significant limitations in terms of task diversity. Currently, the multimodal instruction dataset with the highest number of tasks is vision-flan [1], boasting over 200 tasks. However, this is still considerably fewer than instruction datasets in the NLP community, such as Flan-2022 [2] (1836 tasks) and Ni-v2 [3] (1616 tasks). Therefore, to rapidly increase task diversity, we can consolidate existing multimodal instruction datasets. Designing a unified format is what we consider the foremost requirement for merging these datasets, and this is the significance behind the birth of the UMIT framework.

---

> ### Author Response · Authors · 2023-11-22
> **Author Response for Reviewer 4eXe (Part 2)**
>
> **Q2: The retrieval augmentation in Section 2.4 requires us to have a reasonably large database of training examples to choose from, which may enhance task performance on benchmarks but defeats the purpose of in-context learning in the real world.**
>
> **A1:** Firstly, we think that our work is highly meaningful. Currently, in the multimodal field, there are numerous instruction datasets such as LLaVA, Otter, MultiInstruct, Vision-Flan, etc. However, the tasks included in these datasets are very limited. For example, even Vision-Flan only has 200 tasks, which still represents a significant gap compared to instruction datasets in the NLP community, such as Flan-2022 with over 1800 tasks. Additionally, after visual instruction tuning, the in-context learning capability of MLLMs tends to decline, as evident in Table 2 and Table 3. Therefore, based on these two problems, we propose a unified multimodal instruction format to merge existing datasets, increase task diversity, and, at the same time, maximize the retention of in-context learning ability on traditional visual language tasks.
>
> We strongly agree with your perspective on the significant value of in-context learning in the real world, surpassing its importance in traditional vision-language tasks. However, the focus of our paper still lies in the latter, aiming to enhance the in-context learning ability of MLLMs on traditional vision-language tasks through retrieval methods. There are two reasons for this:
> + Firstly, we argue that, except for GPT-4V, all current MLLMs lack the capability to achieve in-context learning in the real world. For instance, MLLMs like Flamingo [4], OpenFlamingo [5], Kosmos-2 [6], Emu [7], Otter [8] among others, merely showcase their few-shot performance on traditional vision-language tasks. However, there are no other MLLMs demonstrating real-world in-context learning examples as illustrated in Figures 8 — 13 in GPT-4V technical report [9].
> + Secondly, We think that there is currently a lack of a reasonable evaluation method and dataset to effectively measure MLLMs' in-context learning ability in the real world. For example, in Figures 8 to 10 of the GPT-4V technical report [9], GPT-4V accurately judges the speed range of a dashboard through in-context learning, demonstrating the utility of in-context learning. However, we face challenges in quantifying it effectively.
>
> In summary, we believe that exploring the in-context learning ability of MLLMs in the real world still requires further investigation. In our paper, we can only focus on the performance improvement on traditional VL metrics brought about by constructing a unified multimodal instruction format using retrieval method.
>
> **Q3: One ablation experiment I would suggest is to use the task-definition prompt without the instance-level instruction (DEQA).**
>
> **A3:** As we mentioned in our response in **A1**, after adding the "task definition" component, "instance-level" and "question" can be considered as the same component. Therefore, the results of our DEIQA are consistent with what you perceive as DEQA.
>
> **Q4: In equation 5, is the text embedding E_text computed using the question X_q, answer X_a or both?**
>
> **A4:** Thank you for your comments! We are sorry that we didn't make that clear in Section 2.3. We’ve revised the content of Section 2.3.
> In fact, we use text embeddings of both question X_q and answer X_a to retrieve in-context exemplars in training stage. This ensures that we can still retrieve similar in-context examples in tasks with specific questions other than VQA tasks. However, for all evaluations, we randomly select in-context exemplars in testing stage and our method still can achieve promising performances.
>
> [1]  [Xu, Trevor Ashby et al. “Vision-Flan: Scaling Visual Instruction Tuning.”](https://vision-flan.github.io/)
>
> [2]  Longpre, S. et al. “The Flan Collection: Designing Data and Methods for Effective Instruction Tuning.” International Conference on Machine Learning (2023).
>
> [3] Wang, Yizhong et al. “Super-NaturalInstructions: Generalization via Declarative Instructions on 1600+ NLP Tasks.” Conference on Empirical Methods in Natural Language Processing (2022).
>
> [4] Alayrac, Jean-Baptiste et al. “Flamingo: a Visual Language Model for Few-Shot Learning.” ArXiv abs/2204.14198 (2022): n. pag.
>
> [5] Awadalla, Anas et al. “OpenFlamingo: An Open-Source Framework for Training Large Autoregressive Vision-Language Models.” ArXiv abs/2308.01390 (2023): n. pag.
>
> [6] Peng, Zhiliang et al. “Kosmos-2: Grounding Multimodal Large Language Models to the World.” ArXiv abs/2306.14824 (2023): n. pag.
>
> [7] Sun, Quan et al. “Generative Pretraining in Multimodality.” ArXiv abs/2307.05222 (2023): n. pag.
>
> [8] Li, Bo et al. “MIMIC-IT: Multi-Modal In-Context Instruction Tuning.” ArXiv abs/2306.05425 (2023): n. pag.
>
> [9] Yang, Zhengyuan et al. “The Dawn of LMMs: Preliminary Explorations with GPT-4V(ision).” ArXiv abs/2309.17421 (2023): n. pag.

---

> > ### Author Response · Authors · 2023-11-23
> >
> > Dear Reviewer 4eXe,
> >
> > Thank you again for your invaluable insights and observations! We would be immensely grateful if you could share your satisfaction level regarding our response. We will be happy to address any remaining concerns.
> >
> > Sincerely,
> >
> > Paper8964 Authors

---

### Meta-Review · Area_Chair_gytz · 2023-12-11

**Metareview:**

Summary
The paper proposes Unified Multimodal Instruction Tuning (UMIT), aiming to enhance in-context learning in vision and language models. A text-image interleaved instruction dataset is constructed by merging diverse visual instruction datasets in a unified multimodal instruction format with task definition as an additional component in the instruction. The experimental results indicate the advantage of UMIT over in-context learning on various vision-language tasks compared to previous formats.

Strengths:
Investigating the impact of different instruction formats on  in-context learning ability of MLLM is important.
Superior performance to other SOTAs on two MLLM benchmarks.

Weaknesses:
The novelty of this work is limited, where a multimodal instruction format with the component of task definition unifies the instruction across several datasets.
The real-world applicability of UMIT is limited, as it is not possible to effectively describe every task in open domains.
The analysis on the role of task definition in performance increase is not clear.

**Justification For Why Not Higher Score:**

Novelty is limited, less practical in open domains and the improvement brought by task definition is not clear.

**Justification For Why Not Lower Score:**

This work is recommended to be rejected.

---

### Decision · Program_Chairs · 2024-01-16

Reject